# Two-Stage Learning to Defer with Multiple Experts

**Anqi Mao**
Courant Institute
New York, NY 10012
aqmao@cims.nyu.edu

**Christopher Mohri**
Stanford University
Stanford, CA 94305
xmohri@stanford.edu

**Mehryar Mohri**
Google Research & CIMS
New York, NY 10011
mohri@google.com

**Yutao Zhong**
Courant Institute
New York, NY 10012
yutao@cims.nyu.edu

## Abstract

We study a two-stage scenario for learning to defer with multiple experts, which is crucial in practice for many applications. In this scenario, a predictor is derived in a first stage by training with a common loss function such as cross-entropy. In the second stage, a deferral function is learned to assign the most suitable expert to each input. We design a new family of surrogate loss functions for this scenario both in the score-based and the predictor-rejector settings and prove that they are supported by $\mathcal{H}$-consistency bounds, which implies their Bayes-consistency. Moreover, we show that, for a constant cost function, our two-stage surrogate losses are realizable $\mathcal{H}$-consistent. While the main focus of this work is a theoretical analysis, we also report the results of several experiments on CIFAR-10 and SVHN datasets.

## 1 Introduction

Large language models (LLMs) have achieved a remarkable performance on diverse tasks across multiple domains, as reported in recent surveys [Wei et al., 2022, Bubeck et al., 2023]. However, their practical application faces two critical challenges: the occurrence of *hallucinations*, that is the generation of incorrect or misleading content, and an inefficient inference. Leveraging multiple experts can address both issues. To reduce hallucinations, one can refrain from using the original predictor in uncertain instances and defer to one of the more complex and more accurate experts. To enhance efficiency, one can derive models of different sizes distilled from the original complex model and use one of these more streamlined versions, while deferring to the more complex and less efficient ones for suitable contexts. Both problems require assigning each instance to the most suitable expert. This motivates the problem of *learning to defer in the presence of multiple experts*.

The scenario of *single-stage learning to defer* has been studied by several publications, starting with the foundational framework introduced by Cortes, DeSalvo, and Mohri [2016a,b, 2023] for learning to reject and followed by a series of studies on abstention and deferral [Madras et al., 2018, Raghu et al., 2019, Mozannar and Sontag, 2020, Wilder et al., 2021, Pradier et al., 2021, Keswani et al., 2021, Raman and Yee, 2021, Liu et al., 2022, Verma and Nalisnick, 2022, Charusaie et al., 2022, Cao et al., 2022, Verma et al., 2023, Mao et al., 2023f,b,c, Mozannar et al., 2023]. In the single-stage scenario, a predictor and a deferral function are learned simultaneously, with the deferral function determining the best expert assigned to each input instance. However, in practice, a predictor such as an LLM is already available and retraining one in conjunction with a deferral function could be prohibitively costly: depending on its size and the amount of data used, retraining could take several weeks or months. Thus, the single-stage learning to defer scenario and its solutions often do not align with the practical challenges encountered in real-world applications.

37th Conference on Neural Information Processing Systems (NeurIPS 2023).

Alternative post-hoc methods have been proposed to address the learning to defer problem. Okati et al. [2021] proposed an iterative approach optimizing a predictor and a rejector over multiple epochs. Within each epoch, first the predictor is trained on points where its loss is lower than that of a human expert; second, the rejector is fitted to predict which of the predictor or the human expert has a lower loss. Narasimhan et al. [2022] suggested a post-hoc correction to the single-stage learning to defer surrogate losses, specifically the cost-sensitive softmax cross-entropy (CSS) surrogate loss in [Mozannar and Sontag, 2020] and the one-versus-all (OvA) surrogate loss in [Verma and Nalisnick, 2022] for cases where they suffer from underfitting. However, as with the single-stage learning to defer solutions, post-hoc approaches do not apply to scenarios where an existing predictor, pre-trained using a standard classification loss function such as cross-entropy, is already available.

Can we derive a principled algorithm for learning to defer with multiple experts in such scenarios? Which surrogate loss should we adopt and what consistency guarantee can we rely on? This paper deals precisely with these questions.

A key criterion for surrogate losses in learning to defer is Bayes-consistency [Zhang, 2004, Bartlett et al., 2006, Steinwart, 2007], that is minimizing the surrogate loss over the family of measurable functions leads to the minimization of the deferral loss. The surrogate losses proposed in [Mozannar and Sontag, 2020, Verma and Nalisnick, 2022] have been shown to be Bayes-consistent for deferral. However, Bayes-consistency is not relevant in learning tasks since the hypothesis set used, for example that of some family of linear functions or neural networks, never includes all measurable functions. Long and Servedio [2013] proposed a notion of realizable $\mathcal{H}$-consistency, that is consistency associated with a specific hypothesis set in the realizable scenario. Mozannar et al. [2023] recently showed that existing Bayes-consistent surrogate losses in [Mozannar and Sontag, 2020, Verma and Nalisnick, 2022] are not realizable $\mathcal{H}$-consistent for learning with deferral, which can pose significant challenges when learning with a restricted hypothesis set $\mathcal{H}$, even for simple linear models. Instead, they proposed a new surrogate loss that is realizable $\mathcal{H}$-consistent when $\mathcal{H}$ is closed under scaling. However, they also observed that the loss function of Madras et al. [2018], which is not Bayes-consistent, is actually realizable $\mathcal{H}$-consistent. They acknowledged their inability to prove or disprove whether their proposed surrogate loss is Bayes-consistent. Consequently, it has remained an open problem to identify a surrogate loss that is both consistent and realizable-consistent.

In recent work, Verma et al. [2023] proposed the first Bayes-consistent surrogate losses in the scenario of single-stage learning to defer with *multiple experts* [Hemmer et al., 2022, Keswani et al., 2021, Kerrigan et al., 2021, Straitouri et al., 2022, Benz and Rodriguez, 2022]. This scenario is more attractive and significant in applications such as large language models, where multiple models are often available for deferral. However, the surrogate losses proposed by the authors do not benefit from realizable $\mathcal{H}$-consistency, even in the single-expert setting, since they are a straightforward generalization of those of Mozannar and Sontag [2020] and Verma and Nalisnick [2022].

Bayes-consistency, or even realizable $\mathcal{H}$-consistency for a specific hypothesis set $\mathcal{H}$, is an asymptotic property, and provides no guarantee for approximate minimizers since convergence could be arbitrarily slow. More favorable guarantees, known as $\mathcal{H}$-*consistency bounds*, were recently introduced for standard classification settings [Awasthi, Mao, Mohri, and Zhong, 2022b,a]. These guarantees are upper bounds on the target estimation loss expressed in terms of the surrogate estimation loss. They are stronger and more informative guarantees than Bayes-consistency and $\mathcal{H}$-consistency because they are both hypothesis set-specific and non-asymptotic. More recently, Mao et al. [2023b] introduced a new family of surrogate losses and algorithms for the general problem of single-stage learning to defer with multiple experts that benefit from strong $\mathcal{H}$-consistency bounds.

**Our contributions.** We study a two-stage scenario for learning to defer with multiple experts that is crucial in practice for many applications. In this scenario, a predictor is derived in a first stage by training with a common loss function such as cross-entropy. In the second stage, a deferral function is learned to assign the most suitable expert to each input. We design a new family of surrogate loss functions for this scenario both in the *score-based setting* (Section 3) and the *predictor-rejector* setting (Section 4) and prove that they are supported by $\mathcal{H}$-consistency bounds, which implies their Bayes-consistency. Moreover, we show that, for a constant cost function, our two-stage surrogate losses are realizable $\mathcal{H}$-consistent. While the main focus of this work is a theoretical analysis, we also report the results of several experiments on CIFAR-10 and SVHN datasets (Section 5). We give a comprehensive discussion of related work in Appendix A. We begin by providing some basic definitions and notation (Section 2).

## 2 Preliminaries

We consider the standard multi-class classification setting with an input space $\mathcal{X}$ and a set of $n \geq 2$ labels $\mathcal{Y} = [n]$, where we use the notation $[n]$ to denote the set $\{1, \ldots, n\}$. We study the scenario of *learning to defer with multiple experts*, where the label set $\mathcal{Y}$ is augmented with $n_e$ additional labels $\{n + 1, \ldots, n + n_e\}$ corresponding to $n_e$ pre-defined experts $h_1, \ldots, h_{n_e}$. In this scenario, the learner has the option of returning a label $y \in \mathcal{Y}$, which represents the category predicted, or a label $y = n + j$, $j \geq 1$, in which case it is *deferring* to expert $h_j$. This setting is referred to as the *score-based setting* [Mozannar and Sontag, 2020, Cao et al., 2022, Mao et al., 2023f], since the deferral corresponds to extra $n_e$ scoring functions. An alternative setting is the *predictor-rejector setting* [Cortes et al., 2016a, 2023, Mohri et al., 2023, Mao et al., 2023c], where the deferral function is selected from a separate family of functions $\mathcal{R}$. We introduce that setting and include the corresponding results in Section 4 for completeness.

We denote by $\overline{\mathcal{Y}} = [n + n_e]$ the augmented label set and consider a hypothesis set $\mathcal{H}$ of functions mapping from $\mathcal{X} \times \overline{\mathcal{Y}}$ to $\mathbb{R}$. The prediction associated by $h \in \mathcal{H}$ to an input $x \in \mathcal{X}$ is denoted by $\mathsf{h}(x)$ and defined as the element in $\overline{\mathcal{Y}}$ with the highest score, $\mathsf{h}(x) = \mathrm{argmax}_{y \in [n + n_e]} h(x, y)$, with an arbitrary but fixed deterministic strategy for breaking ties. We denote by $\mathcal{H}_{\mathrm{all}}$ the family of all measurable functions.

The *deferral loss function* $\mathsf{L}_{\mathrm{def}}$ is defined as follows for any $h \in \mathcal{H}$ and $(x, y) \in \mathcal{X} \times \mathcal{Y}$:

$$\mathsf{L}_{\mathrm{def}}(h, x, y) = \mathbb{1}_{\mathsf{h}(x) \neq y} \mathbb{1}_{\mathsf{h}(x) \in [n]} + \sum_{j=1}^{n_e} c_j(x, y) \mathbb{1}_{\mathsf{h}(x) = n + j} \tag{1}$$

Thus, the loss incurred coincides with the standard zero-one classification loss when $\mathsf{h}(x)$, the label predicted, is in $\mathcal{Y}$. Otherwise, when $\mathsf{h}(x)$ is equal to $n + j$, the loss incurred is $c_j(x, y)$, the cost of deferring to expert $h_j$. Let $\bar{c}_j(x, y) = 1 - c_j(x, y)$. We will denote by $\underline{c}_j \geq 0$ and $\bar{c}_j \leq 1$ finite lower and upper bounds on the cost $\bar{c}_j$, that is $\bar{c}_j(x, y) \in [\underline{c}_j, \bar{c}_j]$ for all $(x, y) \in \mathcal{X} \times \mathcal{Y}$. There are many possible choices for these costs. Our analysis for Theorem 1, Corollary 2, Theorem 6 is general and requires no assumption other than their boundedness. One natural choice is to define cost $c_j$ as a function of expert $h_j$'s inaccuracy, for example $c_j(x, y) = \alpha_j \mathbb{1}_{\mathsf{h}_j(x) \neq y} + \beta_j$, with $\alpha_j, \beta_j > 0$, where $\mathsf{h}_j(x)$ is the prediction made by $h_j$th for input $x$. Typically, the hyperparameter $\alpha_j$ has two potential values: zero or one. When $\alpha_j$ is set to one, the first term of the formulation pertains to the inaccuracy of expert expert $h_j$. Conversely, with $\alpha_j$ set to zero, the first term vanishes, focusing solely on the inference cost. Theorems 5 and 7 are analyzed under this assumption. The $\beta_j$ in the second term corresponds to the inference cost incurred by expert $h_j$.

Given a distribution $\mathcal{D}$ over $\mathcal{X} \times \mathcal{Y}$, we will denote by $\mathcal{E}_{\mathsf{L}_{\mathrm{def}}}(h)$ the expected deferral loss of a hypothesis $h \in \mathcal{H}$, $\mathcal{E}_{\mathsf{L}_{\mathrm{def}}}(h) = \mathbb{E}_{(x,y) \sim \mathcal{D}}[\mathsf{L}_{\mathrm{def}}(h, x, y)]$, and by $\mathcal{E}_{\mathsf{L}_{\mathrm{def}}}^*(\mathcal{H}) = \inf_{h \in \mathcal{H}} \mathcal{E}_{\mathsf{L}_{\mathrm{def}}}(h)$ its infimum or best-in-class expected loss. We will adopt similar definitions for other loss functions.

Given a hypothesis set $\mathcal{H}$, an $\mathcal{H}$-*consistency bound* [Awasthi et al., 2021a,b, 2022b,a, 2023a,b, Mao et al., 2023h,d,e, Zheng et al., 2023, Mao et al., 2023a,g] for a surrogate loss $\ell_1$ of a target loss function $\ell_2$ is an inequality of the form

$$\forall h \in \mathcal{H}, \ \mathcal{E}_{\ell_2}(h) - \mathcal{E}_{\ell_2}^*(\mathcal{H}) + \mathcal{M}_{\ell_1}(\mathcal{H}) \leq \Gamma\big(\mathcal{E}_{\ell_1}(h) - \mathcal{E}_{\ell_1}^*(\mathcal{H}) + \mathcal{M}_{\ell_1}(\mathcal{H})\big), \tag{2}$$

where $\Gamma \colon \mathbb{R}_+ \to \mathbb{R}_+$ is a non-decreasing function with $\Gamma(0) = 0$ and where $\mathcal{M}_\ell(\mathcal{H})$ is *the minimizability gap* for the hypothesis set $\mathcal{H}$ and loss function $\ell$. $\mathcal{M}_\ell(\mathcal{H})$ is defined as the difference of the best-in-class expected loss and that of the expected pointwise infimum loss: $\mathcal{M}_\ell(\mathcal{H}) = \mathcal{E}_\ell^*(\mathcal{H}) - \mathbb{E}_x\big[\inf_{h \in \mathcal{H}} \mathbb{E}_{y|x}[\ell(h, x, y)]\big]$. By the super-additivity of the infimum, the minimizability gap is always non-negative. The minimizability gap vanishes when the best-in-class error $\mathcal{E}_\ell^*(\mathcal{H})$ coincides with the Bayes error $\mathcal{E}_\ell^*(\mathcal{H}_{\mathrm{all}})$, in particular when $\mathcal{H} = \mathcal{H}_{\mathrm{all}}$ [Awasthi et al., 2022a,b].

Thus, the $\mathcal{H}$-consistency bound (2) relates the minimization of the estimation error for the surrogate loss $\ell_1$ to that of the target loss $\ell_2$ in a quantitative way. It is a stronger and more informative guarantee than Bayes-consistency which implies Bayes-consistency, as can be seen by setting $\mathcal{H} = \mathcal{H}_{\mathrm{all}}$.

Table 1: Common surrogate losses in standard multi-class classification.

| Name | Formulation |
|------|-------------|
| Sum exponential loss | $\ell_{\exp}(\overline{h}, x, y) = \sum_{y' \neq y} e^{\overline{h}(x,y') - \overline{h}(x,y)}.$ |
| Multinomial logistic loss | $\ell_{\log}(\overline{h}, x, y) = \log\left(\sum_{y' \in \mathcal{Y} \cup \{0\}} e^{\overline{h}(x,y') - \overline{h}(x,y)}\right).$ |
| Generalized cross-entropy loss | $\ell_{\mathrm{gce}}(\overline{h}, x, y) = \frac{1}{\alpha}\left[1 - \left[\frac{e^{\overline{h}(x,y)}}{\sum_{y' \in \mathcal{Y} \cup \{0\}} e^{\overline{h}(x,y')}}\right]^{\alpha}\right], \alpha \in (0,1).$ |
| Mean absolute error loss | $\ell_{\mathrm{mae}}(\overline{h}, x, y) = 1 - \frac{e^{\overline{h}(x,y)}}{\sum_{y' \in \mathcal{Y} \cup \{0\}} e^{\overline{h}(x,y')}}.$ |

# 3 Two-stage $\mathcal{H}$-consistent surrogate loss

In this section, we consider an important *two-stage* scenario for learning to defer with multiple experts. This is a critical scenario in practice for many applications where a predictor is already available, as a result of training with a loss function $\ell$ supported by $\mathcal{H}$-consistency bounds, such as the logistic loss (first stage). The logistic loss coincides with the cross-entropy loss when a softmax activation is applied to the output of a neural network. The problem then consists of learning a deferral function (second stage) to assign the most suitable expert to each input instance.

We first design a new family of surrogate losses for this *two-stage* scenario (Section 3.1). Next, we show that our surrogate losses benefit from $\mathcal{H}$-consistency bounds (Section 3.2). As a by-product, we prove $\overline{\mathcal{H}}$-consistency bounds in standard multi-class classification, where $\overline{\mathcal{H}}$ denotes hypothesis sets with a fixed scoring function (Section 3.3). These bounds have not been studied before and can be of independent interest in other consistency studies. Moreover, we show that, for a constant cost function, our two-stage surrogate losses are realizable $\mathcal{H}$-consistent (Section 3.4).

## 3.1 General surrogate losses

A hypothesis set $\mathcal{H}$ of functions mapping from $\mathcal{X} \times [n + n_e]$ to $\mathbb{R}$ can be decomposed as $\mathcal{H} = \mathcal{H}_p \times \mathcal{H}_d$, where $\mathcal{H}_p$ denotes the hypothesis set spanned by the first $n$ scores, used for prediction, and $\mathcal{H}_d$ the hypothesis set spanned by the final $n_e$ scores, used for deferral. Thus, any $h \in \mathcal{H}$ can be written as a pair $h = (h_p, h_d)$ with $h_p \in \mathcal{H}_p$ and $h_d \in \mathcal{H}_d$.

Let $\ell$ be a surrogate loss for standard multi-class classification with $n$ classes. We consider the following two-stage scenario: in the first stage, $h_p$ is learned using the surrogate loss $\ell_1$; in the second stage, $h_d$ is learned using a surrogate loss $\mathsf{L}_{h_p}$ that depends on the prediction function $h_p$ learned in the first stage.

To any $h_d \in \mathcal{H}_d$, we associate a hypothesis $\overline{h}_d$ defined over $(n_e + 1)$ classes $\{0, 1, \ldots, n_e\}$ by $\overline{h}_d(x, 0) = \max_{y \in \mathcal{Y}} h_p(x, y)$, that is the maximal score assigned by $h_p$ to its predicted label, and $\overline{h}_d(x, j) = h_d(x, j)$ for $j \in [n_e]$. We can then define our suggested surrogate loss for the second stage as follows:

$$\mathsf{L}_{h_p}(h_d, x, y) = \mathbb{1}_{\mathsf{h}_\mathsf{p}(x) = y} \ell_2(\overline{h}_d, x, 0) + \sum_{j=1}^{n_e} \overline{c}_j(x, y) \ell_2(\overline{h}_d, x, j), \tag{3}$$

where $\ell_2(\overline{h}_d, x, j)$ is a surrogate loss for standard multi-class classification with $(n_e + 1)$ categories $\{0, 1, \ldots, n_e\}$. Intuitively, the indicator term $\mathbb{1}_{h(x) \neq n+j}$ in the deferral loss (1) penalizes $h_d(x, j)$ when it has a small value. Similarly, for a standard surrogate loss $\ell_2(\overline{h}_d, x, j)$ such as the logistic loss, it penalizes $\overline{h}_d(x, j)$ when it has a small value as well. In Table 2, we present a summary of examples of such second-stage surrogate losses, where $\ell_2$ is selected from common surrogate losses in standard multi-class classification defined in Table 1. A detailed derivation is presented in Appendix B.

From the point of view of the second stage, $x \mapsto \overline{h}_d(x, 0) = \max_{y \in \mathcal{Y}} h_p(x, y)$ is a fixed function. We will denote by $\overline{\mathcal{H}}_d$ the family of hypotheses $\overline{h}_d \colon \mathcal{X} \times \{0, 1, \ldots, n_e\} \to \mathbb{R}$ whose first scoring function, $\overline{h}_d(\cdot, 0)$, is fixed and not to be learned in the second stage.

Our formulation bears some similarity with the design of a surrogate loss function for rejectors in [Cortes et al., 2016a, 2023] for learning with rejection in binary classification, where the cost is a

constant. However, our surrogate loss is tailored to accommodate a general cost function depending on both $x$ and $y$ for deferral, in contrast with a constant one, and it allows for multiple deferral options, as opposed to only one rejection option.

### 3.2 $\mathcal{H}$-consistency bounds for two-stage surrogate losses

In this section, we provide strong guarantees for two-stage surrogate losses, provided that the first-stage loss function $\ell_1$ admits an $\mathcal{H}_p$-consistency bound, and the second-stage surrogate $\ell_2$ admits an $\overline{\mathcal{H}}_d$-consistency bound.

**Theorem 1** ($\mathcal{H}$-**consistency bounds for score-based two-stage surrogates**). *Assume that $\ell_1$ admits an $\mathcal{H}_p$-consistency bound and $\ell_2$ admits an $\overline{\mathcal{H}}_d$-consistency bound with respect to the multi-class zero-one classification loss $\ell_{0-1}$ respectively. Thus, there are non-decreasing concave functions $\Gamma_1$ and $\Gamma_2$ such that, for all $h_p \in \mathcal{H}_p$ and $\overline{h}_d \in \overline{\mathcal{H}}_d$, we have*

$$\mathcal{E}_{\ell_{0-1}}(h_p) - \mathcal{E}_{\ell_{0-1}}^*(\mathcal{H}_p) + \mathcal{M}_{\ell_{0-1}}(\mathcal{H}_p) \leq \Gamma_1\big(\mathcal{E}_{\ell_1}(h_p) - \mathcal{E}_{\ell_1}^*(\mathcal{H}_p) + \mathcal{M}_{\ell_1}(\mathcal{H}_p)\big)$$

$$\mathcal{E}_{\ell_{0-1}}(\overline{h}_d) - \mathcal{E}_{\ell_{0-1}}^*(\overline{\mathcal{H}}_d) + \mathcal{M}_{\ell_{0-1}}(\overline{\mathcal{H}}_d) \leq \Gamma_2\big(\mathcal{E}_{\ell_2}(\overline{h}_d) - \mathcal{E}_{\ell_2}^*(\overline{\mathcal{H}}_d) + \mathcal{M}_{\ell_2}(\overline{\mathcal{H}}_d)\big).$$

*Then, the following holds for all $h \in \mathcal{H}$:*

$$\mathcal{E}_{\mathsf{L}_{\mathrm{def}}}(h) - \mathcal{E}_{\mathsf{L}_{\mathrm{def}}}^*(\mathcal{H}) + \mathcal{M}_{\mathsf{L}_{\mathrm{def}}}(\mathcal{H})$$

$$\leq \Gamma_1\big(\mathcal{E}_{\ell_1}(h_p) - \mathcal{E}_{\ell_1}^*(\mathcal{H}_p) + \mathcal{M}_{\ell_1}(\mathcal{H}_p)\big) + \left(1 + \sum_{j=1}^{n_e} \overline{c}_j\right)\Gamma_2\left(\frac{\mathcal{E}_{\mathsf{L}_{h_p}}(h_d) - \mathcal{E}_{\mathsf{L}_{h_p}}^*(\mathcal{H}_d) + \mathcal{M}_{\mathsf{L}_{h_p}}(\mathcal{H}_d)}{\sum_{j=1}^{n_e} \underline{c}_j}\right).$$

*Furthermore, constant factors $\left(1 + \sum_{j=1}^{n_e} \overline{c}_j\right)$ and $\frac{1}{\sum_{j=1}^{n_e} \underline{c}_j}$ can be removed when $\Gamma_2$ is linear.*

The proof is given in Appendix D. It consists of expressing the conditional regret of the deferral loss as the sum of two regrets, first by minimizing $h_d$ for a fixed $h_p$ and then by minimizing $h_p$. Subsequently, we show how each regret can be upper-bounded in terms of the conditional regret of each stage's surrogate loss, leveraging the $\mathcal{H}_p$-consistency bound of $\ell_1$ and $\overline{\mathcal{H}}_d$-consistency bound of $\ell_2$ with respect to the zero-one loss. This, in conjunction with the concavity of functions $\Gamma_1$ and $\Gamma_2$, establishes our $\mathcal{H}$-consistency bounds.

Thus, the theorem provides a strong guarantee for the two-stage surrogate losses. A specific instance of Theorem 1 holds for the case where $\mathcal{E}_{\ell_1}^*(\mathcal{H}_p) = \mathcal{E}_{\ell_1}^*(\mathcal{H}_{\mathrm{all}})$ and $\mathcal{E}_{\mathsf{L}_{h_p}}^*(\mathcal{H}_d) = \mathcal{E}_{\mathsf{L}_{h_p}}^*(\mathcal{H}_{\mathrm{all}})$, ensuring that the Bayes-error coincides with the best-in-class error and, consequently, $\mathcal{M}_{\ell_1}(\mathcal{H}_p) = \mathcal{M}_{\mathsf{L}_{h_p}}(\mathcal{H}_d) = 0$. Given Theorem 1 and the non-negativity property of $\mathcal{M}_{\mathsf{L}_{\mathrm{def}}}(\mathcal{H})$, we can derive the following corollary.

**Corollary 2.** *Assume that $\ell$ satisfies the same assumption as in Theorem 1. Then, for all $h \in \mathcal{H}$ and any distribution such that $\mathcal{E}_{\ell_1}^*(\mathcal{H}_p) = \mathcal{E}_{\ell_1}^*(\mathcal{H}_{\mathrm{all}})$ and $\mathcal{E}_{\mathsf{L}_{h_p}}^*(\mathcal{H}_d) = \mathcal{E}_{\mathsf{L}_{h_p}}^*(\mathcal{H}_{\mathrm{all}})$, we have*

$$\mathcal{E}_{\mathsf{L}_{\mathrm{def}}}(h) - \mathcal{E}_{\mathsf{L}_{\mathrm{def}}}^*(\mathcal{H}) \leq \Gamma_1\big(\mathcal{E}_{\ell_1}(h_p) - \mathcal{E}_{\ell_1}^*(\mathcal{H}_p)\big) + \left(1 + \sum_{j=1}^{n_e} \overline{c}_j\right)\Gamma_2\left(\frac{\mathcal{E}_{\mathsf{L}_{h_p}}(h_d) - \mathcal{E}_{\mathsf{L}_{h_p}}^*(\mathcal{H}_d)}{\sum_{j=1}^{n_e} \underline{c}_j}\right),$$

*where the constant factors $\left(1 + \sum_{j=1}^{n_e} \overline{c}_j\right)$ and $\frac{1}{\sum_{j=1}^{n_e} \underline{c}_j}$ can be removed when $\Gamma_2$ is linear.*

Corollary 2 implies that when the estimation error of the first-stage surrogate loss, $\mathcal{E}_{\ell_1}(h_p) - \mathcal{E}_{\ell_1}^*(\mathcal{H}_p)$, is reduced to $\epsilon_1$, and the estimation error of the second-stage surrogate loss, $\mathcal{E}_{\mathsf{L}_{h_p}}(h_d) - \mathcal{E}_{\mathsf{L}_{h_p}}^*(\mathcal{H}_d)$, is reduced to $\epsilon_2$, the estimation error of the deferral loss, $\mathcal{E}_{\mathsf{L}_{\mathrm{def}}}(h) - \mathcal{E}_{\mathsf{L}_{\mathrm{def}}}^*(\mathcal{H})$, is upper-bound by

$$\Gamma_1(\epsilon_1) + \left(1 + \sum_{j=1}^{n_e} \overline{c}_j\right)\Gamma_2\left(\frac{\epsilon_2}{\sum_{j=1}^{n_e} \underline{c}_j}\right).$$

The common surrogate losses mentioned earlier all satisfy the first-stage requirement; however, it was unclear if they would meet the second-stage criterion since the $\overline{\mathcal{H}}_d$-consistency bound is for hypothesis sets $\overline{\mathcal{H}}_d$ with a fixed first scoring function. This has not been previously studied in the literature. In the next section, we prove for the first time that common multi-class surrogate losses,

Table 2: Examples for score-based second-stage surrogate losses (3).

| $\ell_2$ | $\mathsf{L}_{h_p}$ |
|---|---|
| $\ell_{\exp}$ | $\mathbb{1}_{\mathsf{h}_{\mathsf{p}}(x)=y} \sum_{i=1}^{n_e} e^{h(x,n+i)-\max_{y\in\mathcal{Y}} h(x,y)} + \sum_{j=1}^{n_e} \overline{c}_j(x,y)\big[\sum_{i=1,i\neq j}^{n_e} e^{h(x,n+i)-h(x,n+j)} + e^{\max_{y\in\mathcal{Y}} h(x,y)-h(x,n+j)}\big]$ |
| $\ell_{\log}$ | $-\mathbb{1}_{\mathsf{h}_{\mathsf{p}}(x)=y} \log\Big(\frac{e^{\max_{y\in\mathcal{Y}} h(x,y)}}{e^{\max_{y\in\mathcal{Y}} h(x,y)}+\sum_{i=1}^{n_e} e^{h(x,n+i)}}\Big) - \sum_{j=1}^{n_e} \overline{c}_j(x,y) \log\Big(\frac{e^{h(x,n+j)}}{e^{\max_{y\in\mathcal{Y}} h(x,y)}+\sum_{i=1}^{n_e} e^{h(x,n+i)}}\Big)$ |
| $\ell_{\mathrm{gce}}$ | $\mathbb{1}_{\mathsf{h}_{\mathsf{p}}(x)=y} \frac{1}{\alpha}\Big[1-\Big[\frac{e^{\max_{y\in\mathcal{Y}} h(x,y)}}{e^{\max_{y\in\mathcal{Y}} h(x,y)}+\sum_{i=1}^{n_e} e^{h(x,n+i)}}\Big]^{\alpha}\Big] + \sum_{j=1}^{n_e} \overline{c}_j(x,y) \frac{1}{\alpha}\Big[1-\Big[\frac{e^{h(x,n+j)}}{e^{\max_{y\in\mathcal{Y}} h(x,y)}+\sum_{i=1}^{n_e} e^{h(x,n+i)}}\Big]^{\alpha}\Big]$ |
| $\ell_{\mathrm{mae}}$ | $\mathbb{1}_{\mathsf{h}_{\mathsf{p}}(x)=y}\Big[1-\frac{e^{\max_{y\in\mathcal{Y}} h(x,y)}}{e^{\max_{y\in\mathcal{Y}} h(x,y)}+\sum_{i=1}^{n_e} e^{h(x,n+i)}}\Big] + \sum_{j=1}^{n_e} \overline{c}_j(x,y)\Big[1-\frac{e^{h(x,n+j)}}{e^{\max_{y\in\mathcal{Y}} h(x,y)}+\sum_{i=1}^{n_e} e^{h(x,n+i)}}\Big]$ |

such as the logistic loss, satisfy this requirement and can be incorporated into both the first and second stage. Hence, based on [Mao et al., 2023h, Theorem 1] and Theorem 3 in Section 3.3, when using logistic loss in both stages, the concave functions are $\Gamma_1(t) = \Gamma_2(t) = \sqrt{2t}$, and thus Corollary 2 yields the following $\mathcal{H}$-consistency bound:

$$\mathcal{E}_{\mathsf{L}_{\mathrm{def}}}(h) - \mathcal{E}^*_{\mathsf{L}_{\mathrm{def}}}(\mathcal{H}) \leq \sqrt{2}\big[\mathcal{E}_{\ell_1}(h_p) - \mathcal{E}^*_{\ell_1}(\mathcal{H}_p)\big]^{\frac{1}{2}} + \sqrt{2}\Big[1 + \sum_{j=1}^{n_e} \overline{c}_j\Big]\Big[\frac{\mathcal{E}_{\mathsf{L}_{h_p}}(h_d) - \mathcal{E}^*_{\mathsf{L}_{h_p}}(\mathcal{H}_d)}{\sum_{j=1}^{n_e} \underline{c}_j}\Big]^{\frac{1}{2}}.$$

In particular, the bound implies the Bayes-consistency of the two-stage surrogate loss when $\ell_1 = \ell_2 = \ell_{\log}$. Similarly, for other choices of $\ell_1$ and $\ell_2$ defined in Table 1, the two-stage surrogate loss benefits from an $\mathcal{H}$-consistency bound and is also Bayes-consistent.

### 3.3 $\overline{\mathcal{H}}$-consistency bounds for standard surrogate loss functions

In this section, we seek to derive $\overline{\mathcal{H}}$-consistency bounds for common surrogate losses defined in Table 1 in the standard multi-class classification scenario. Recall that the first scoring function of hypotheses in $\overline{\mathcal{H}}_d$ is the function $\max_{y\in\mathcal{Y}} h_p(\cdot, y)$. Here, for any given function $\lambda$ mapping from $\mathcal{X}$ to $\mathbb{R}$, we define the hypothesis set $\overline{\mathcal{H}}$ augmented by $\lambda$ in a similar way, that is to any $h \in \mathcal{H}$ we associate a hypothesis $\overline{h} \in \overline{\mathcal{H}}$ defined by $\overline{h}(x,0) = \lambda(x)$ and $\overline{h}(x,j) = h(x,j)$ for $j \geq 1$. These $\overline{\mathcal{H}}$-consistency bounds offer strong guarantees when the loss functions in Table 1 are used in the second stage of the two-stage learning to defer surrogate losses (3) instantiated in Table 2. We believe that these results are of independent interest and can admit other applications in the study of $\mathcal{H}$-consistency bounds. As with [Mao et al., 2023h], we assume that the hypothesis set $\mathcal{H}$ is *symmetric and complete*. A hypothesis set is said to be *symmetric* if there exists a family $\mathcal{F}$ of functions $f$ mapping from $\mathcal{X}$ to $\mathbb{R}$ such that $\{[h(x,1),\ldots,h(x,n)]: h \in \mathcal{H}\} = \{[f_1(x),\ldots,f_n(x)]: f_1,\ldots,f_n \in \mathcal{F}\}$, for any $x \in \mathcal{X}$. A hypothesis set $\mathcal{H}$ is said to be *complete* if the set of scores it generates spans $\mathbb{R}$, that is, $\{h(x,y): h \in \mathcal{H}\} = \mathbb{R}$, for any $(x,y) \in \mathcal{X} \times \mathcal{Y}$.

Note that for a symmetric and complete $\mathcal{H}$, the associated $\overline{\mathcal{H}}$ is not symmetric and complete. Therefore, the proof of Mao et al. [2023h] cannot be generalized to our setting. Our proofs are presented in Appendix E. We give a new method for upper bounding the conditional regret of the zero-one loss by that of a surrogate loss. To achieve this, we upper bound the minimal conditional surrogate loss by the conditional loss of a carefully constructed hypothesis in $\overline{\mathcal{H}}$ denoted by $\overline{h}_\mu$. The resulting softmax $\mathcal{S}_\mu$ of this hypothesis only differs from the original softmax $\mathcal{S}$ corresponding to $\overline{h}$ on exactly two of the labels.

**Theorem 3** ($\overline{\mathcal{H}}$-consistency bounds). *Assume that $\mathcal{H}$ is symmetric and complete. Then, for any function $\lambda$ mapping from $\mathcal{X}$ to $\mathbb{R}$, hypothesis $\overline{h}$ in the associated hypothesis set $\overline{\mathcal{H}}$ and any distribution, the following inequality holds:*

$$\mathcal{E}_{\ell_{0-1}}(\overline{h}) - \mathcal{E}^*_{\ell_{0-1}}(\overline{\mathcal{H}}) \leq \Gamma\big(\mathcal{E}_\ell(\overline{h}) - \mathcal{E}^*_\ell(\overline{\mathcal{H}}) + \mathcal{M}_\ell(\overline{\mathcal{H}})\big) - \mathcal{M}_{\ell_{0-1}}(\overline{\mathcal{H}}),$$

*where $\Gamma(t) = \sqrt{2t}$ for $\ell = \ell_{\log}$ or $\ell_{\exp}$; $\Gamma(t) = \sqrt{2(n+1)^\alpha t}$ for $\ell = \ell_{\mathrm{gce}}$; and $\Gamma(t) = (n+1)t$ for $\ell = \ell_{\mathrm{mae}}$.*

Let us underscore that our proof technique is novel and distinct from the approach used in [Mao et al., 2023h]. Their method is tailored for hypothesis sets where each score can span across $\mathbb{R}$. This is not applicable in our context where the hypothesis set adheres to a predefined scoring function.

In their proof, to set an upper bound on the estimation error of the zero-one loss using that of the surrogate loss, they select an auxiliary function $\overline{h}_\mu$ for any hypothesis $h$. This function is contingent on the distinct scores of $h$. Subsequently, the authors choose an optimal $\mu$ to set these bounds. Nevertheless, if any of $h$'s scores are fixed, an optimal $\mu$ does not exist, preventing the establishment of a meaningful bound. Instead, our new proof method overcomes this limitation by choosing $\overline{h}_\mu$ based on the softmax, as the softmax corresponding to the label zero can still vary due to the influence of changes in other scores, even when the scoring function on label zero is fixed.

### 3.4 Realizable $\mathcal{H}$-consistency

Recently, Mozannar et al. [2023] showed that even in the straightforward single-expert setting, existing Bayes-consistent single-stage surrogate losses [Mozannar and Sontag, 2020, Verma and Nalisnick, 2022] are not *realizable $\mathcal{H}$-consistent* [Long and Servedio, 2013, Zhang and Agarwal, 2020] for learning with deferral. This can pose significant challenges when learning with a restricted hypothesis set $\mathcal{H}$, even for simple linear models. Instead, they proposed a new surrogate loss that is realizable $\mathcal{H}$-consistent when $\mathcal{H}$ is *closed under scaling*, meaning that it satisfies the condition $h \in \mathcal{H} \Rightarrow \tau h \in \mathcal{H}$ for all $\tau$ in the set of real numbers. However, they stated that they could not prove or disprove whether their proposed surrogate loss is Bayes-consistent. Consequently, it has become crucial to identify a surrogate loss that is both consistent and realizable-consistent, which has remained an open problem.

**Definition 4** (**Realizable $\mathcal{H}$-consistency**). *A surrogate loss $\mathsf{L}$ is considered a* realizable $\mathcal{H}$-consistent *loss function for the deferral loss $\mathsf{L}_{\mathrm{def}}$ if, for any distribution that is $\mathcal{H}$-realizable, that is, there exists a zero loss solution $h^* \in \mathcal{H}$ with $\mathcal{E}_{\mathsf{L}_{\mathrm{def}}}(h^*) = 0$, optimizing the surrogate loss results in obtaining the zero-error solution:*

$$\mathcal{E}_{\mathsf{L}}(h_n) - \mathcal{E}_{\mathsf{L}}^*(\mathcal{H}) \xrightarrow{n \to +\infty} 0 \implies \mathcal{E}_{\mathsf{L}_{\mathrm{def}}}(h_n) - \mathcal{E}_{\mathsf{L}_{\mathrm{def}}}^*(\mathcal{H}) \xrightarrow{n \to +\infty} 0.$$

In the following result, we show that our two-stage surrogate losses are realizable $\mathcal{H}$-consistent. Combined with their Bayes-consistency properties, which have already been established in Section 3.2, we effectively find surrogate losses that are both Bayes-consistent and realizable consistent in the multi-expert setting, including the single-expert setting as a special case. For simplicity, here, we study the case where $\ell_1 = \ell_2 = \ell_{\log}$, a similar proof holds for other choices of $\ell_1$ and $\ell_2$ defined in Table 1. The proof is included in Appendix F.

**Theorem 5** (**Realizable $\mathcal{H}$-consistency for score-based two-stage surrogates**). *Assume that $\mathcal{H}$ is closed under scaling and $c_j(x, y) = \beta_j, \forall (x, y) \in \mathcal{X} \times \mathcal{Y}$. Let $\ell_1$ and $\ell_2$ be the logistic loss. Let $\hat{h}_p$ be the minimizer of $\mathcal{E}_{\ell_1}$ and $\hat{h}_d$ be the minimizer of $\mathcal{E}_{\mathsf{L}_{\hat{h}_p}}$ such that $\mathcal{E}_{\mathsf{L}_{\hat{h}_p}}(\hat{h}_d) = \min_h \mathcal{E}_{\mathsf{L}_{\hat{h}_p}}(h_d)$. Then, the following equality holds for any $(\mathcal{H}, \mathcal{R})$-realizable distribution,*

$$\mathcal{E}_{\mathsf{L}_{\mathrm{def}}}(\hat{h}) = 0, \text{ where } \hat{h} = (\hat{h}_p, \hat{h}_d).$$

Theorem 5 suggests that when the estimation error of the first-stage surrogate loss, $\mathcal{E}_{\ell_1}(h_p^n) - \mathcal{E}_{\ell_1}^*(\mathcal{H}_p) \xrightarrow{n \to +\infty} 0$, and the estimation error of the second-stage surrogate loss, $\mathcal{E}_{\mathsf{L}_{h_p}}(h_d^n) - \mathcal{E}_{\mathsf{L}_{h_p}}^*(\mathcal{H}_d) \xrightarrow{n \to +\infty} 0$, the estimation error of the deferral loss, $\mathcal{E}_{\mathsf{L}_{\mathrm{def}}}(h^n) - \mathcal{E}_{\mathsf{L}_{\mathrm{def}}}^*(\mathcal{H}) \xrightarrow{n \to +\infty} 0$. This result demonstrates that our two-stage surrogate losses are not only Bayes-consistent, but also realizable $\mathcal{H}$-consistent when only the inference cost ($\beta_j$) exists.

## 4 Predictor-rejector setting

The results of the previous sections were all given for the score-based setting. We note that another popular setting in learning with deferral/abstention is the *predictor-rejector setting* [Cortes et al., 2016a, 2023], where the deferral corresponds to a separate function $\mathcal{R}$ instead of extra scores. For completeness, we introduce this setting as well. Here too, we design a family of two-stage surrogate losses benefiting from both $(\mathcal{H}, \mathcal{R})$-consistency bounds and realizable consistency. For simplicity, we overload the notation as with score-based setting based on the context.

Let $\mathcal{H}$ be a hypothesis set of prediction functions mapping from $\mathcal{X} \times \mathcal{Y}$ to $\mathbb{R}$. The label predicted for $x \in \mathcal{X}$ using a hypothesis $h \in \mathcal{H}$ is denoted by $\mathsf{h}(x)$ and defined as one with the highest score,

Table 3: Examples for predictor-rejector second-stage surrogate losses (5).

| $\ell_2$ | $\mathsf{L}_{h_p}$ |
|---|---|
| $\ell_{\exp}$ | $\mathbb{1}_{\mathsf{h}(x)=y}\sum_{i=1}^{n_e}e^{-r_i(x)} + \sum_{j=1}^{n_e}\bar{c}_j(x,y)\Big[\sum_{i=1,i\neq j}^{n_e}e^{r_j(x)-r_i(x)} + e^{r_j(x)}\Big]$ |
| $\ell_{\log}$ | $-\mathbb{1}_{\mathsf{h}(x)=y}\log\Big(\frac{1}{1+\sum_{i=1}^{n_e}e^{-r_i(x)}}\Big) - \sum_{j=1}^{n_e}\bar{c}_j(x,y)\log\Big(\frac{e^{-r_j(x)}}{1+\sum_{i=1}^{n_e}e^{-r_i(x)}}\Big)$ |
| $\ell_{\mathrm{gce}}$ | $\mathbb{1}_{\mathsf{h}(x)=y}\frac{1}{\alpha}\Big[1-\Big[\frac{1}{1+\sum_{i=1}^{n_e}e^{-r_i(x)}}\Big]^{\alpha}\Big] + \sum_{j=1}^{n_e}\bar{c}_j(x,y)\frac{1}{\alpha}\Big[1-\Big[\frac{e^{-r_j(x)}}{1+\sum_{i=1}^{n_e}e^{-r_i(x)}}\Big]^{\alpha}\Big]$ |
| $\ell_{\mathrm{mae}}$ | $\mathbb{1}_{\mathsf{h}(x)=y}\Big[1-\frac{1}{1+\sum_{i=1}^{n_e}e^{-r_i(x)}}\Big] + \sum_{j=1}^{n_e}\bar{c}_j(x,y)\Big[1-\frac{e^{-r_j(x)}}{1+\sum_{i=1}^{n_e}e^{-r_i(x)}}\Big]$ |

$\mathsf{h}(x) = \operatorname{argmax}_{y\in\mathcal{Y}}h(x,y)$, with an arbitrary but fixed deterministic strategy for breaking ties. Let $\mathcal{R}$ be a family of *deferring* functions mapping from $\mathcal{X}$ to $\mathbb{R}^{n_e}$, where $n_e$ is the number of experts. A deferral $r = (r_1, \ldots, r_{n_e}) \in \mathcal{R}$ is used to defer the prediction on input $x$ to the $j$th expert $h_j$ if $r_j(x) \leq 0$ and $r_j(x) < \min_{i=1,i\neq j}^{n_e} r_i(x)$, in which case a cost $c_j(x,y) = 1 - \bar{c}_j(x,y) \in [1-\bar{c}_j, 1-\underline{c}_j]$ is incurred with $0 < \underline{c}_j \leq \bar{c}_j \leq 1$. A natural choice of the cost is $c_j(x,y) = \alpha_j\mathbb{1}_{\mathsf{h}_j(x)\neq y} + \beta_j$, where $\alpha_j, \beta_j > 0$ and $\mathsf{h}_j$ is the prediction of the $j$th expert. The $\beta_j$ in the second term corresponds to the inference cost incurred by expert $h_j$. Let $r_0 = 0$ and define $\mathsf{r}(x) = 0$ if $r_0(x) < \min_{j\in[n_e]} r_j(x)$; otherwise, $\mathsf{r}(x) = \operatorname{argmin}_{j\in[n_e]} r_j(x)$, with an arbitrary but fixed deterministic strategy for breaking ties. The *learning to defer loss* $\mathsf{L}_{\mathrm{def}}$ with $n_e$ experts is defined as follows for any $(h,r) \in \mathcal{H} \times \mathcal{R}$ and $(x,y) \in \mathcal{X} \times \mathcal{Y}$:

$$\mathsf{L}_{\mathrm{def}}(h,r,x,y) = \mathbb{1}_{\mathsf{h}(x)\neq y}\mathbb{1}_{\mathsf{r}(x)=0} + \sum_{j=1}^{n_e} c_j(x,y)\mathbb{1}_{\mathsf{r}(x)=j}. \tag{4}$$

Given a distribution $\mathcal{D}$ over $\mathcal{X} \times \mathcal{Y}$, we will denote by $\mathcal{E}_{\mathsf{L}_{\mathrm{def}}}(h,r)$ the expected deferral loss of a predictor $h \in \mathcal{H}$ and a deferral $r \in \mathcal{R}$, $\mathcal{E}_{\mathsf{L}_{\mathrm{def}}}(h,r) = \mathbb{E}_{(x,y)\sim\mathcal{D}}[\mathsf{L}_{\mathrm{def}}(h,r,x,y)]$, and by $\mathcal{E}^*_{\mathsf{L}_{\mathrm{def}}}(\mathcal{H},\mathcal{R}) = \inf_{h\in\mathcal{H},r\in\mathcal{R}} \mathcal{E}_{\mathsf{L}_{\mathrm{def}}}(h,r)$ its infimum or best-in class expected loss. We will adopt similar definitions for other loss functions. We denote by $\mathcal{M}_{\mathsf{L}}(\mathcal{H},\mathcal{R}) = \mathcal{E}^*_{\mathsf{L}}(\mathcal{H},\mathcal{R}) - \mathbb{E}_x[\inf_{h\in\mathcal{H},r\in\mathcal{R}} \mathbb{E}_{y|x}[\mathsf{L}(h,r,x,y)]]$ the minimizability gap for hypothesis sets $(\mathcal{H},\mathcal{R})$ and a loss function $\mathsf{L}$.

Let $\ell_1$ be a surrogate loss for standard multi-class classification with $n$ classes. We consider the following two-stage scenario: in the first stage, a predictor $h$ is learned using the surrogate loss $\ell_1$; in the second stage, $r$ is learned using a surrogate loss $\mathsf{L}_h$ that depends on the prediction function $h$ learned in the first stage.

To any $r \in \mathcal{R}$, we associate a hypothesis $\bar{r}$ defined over $(n_e + 1)$ classes $\{0, 1, \ldots, n_e\}$ by $\bar{r}(x,0) = 0$, that is zero scoring function, and $\bar{r}(x,j) = -r_j(x)$ for $j \in [n_e]$. We can then define our suggested surrogate loss for the second stage:

$$\mathsf{L}_h(r,x,y) = \mathbb{1}_{\mathsf{h}(x)=y}\ell_2(\bar{r},x,0) + \sum_{j=1}^{n_e}\bar{c}_j(x,y)\ell_2(\bar{r},x,j). \tag{5}$$

Here, $\ell_2(\bar{r},x,j)$ is a surrogate loss for standard multi-class classification with $(n_e + 1)$ categories $\{0, 1, \ldots, n_e\}$. Intuitively, the indicator term $\mathbb{1}_{\mathsf{r}(x)\neq j}$ in the deferral loss penalizes $r_j(x)$ when it has a large value. However, a standard surrogate loss $\ell_2(\bar{r},x,j)$ such as the logistic loss penalizes $\bar{r}(x,j)$ when it has a small value. This is why we use a negative sign in the definition of $\bar{r}$ to maintain consistency between the definitions of $\mathsf{L}_h$ and $\mathsf{L}_{\mathrm{def}}$. In Table 3, we present a summary of examples of such second-stage surrogate losses, where $\ell_2$ is selected from common surrogate losses in standard multi-class classification defined in Table 1. A detailed derivation is presented in Appendix C.

From the point of view of the second stage, we will denote by $\overline{\mathcal{R}}$ the family of hypotheses $\bar{r}: \mathcal{X} \times \{0, 1, \ldots, n_e\} \to \mathbb{R}$ whose first scoring function, $\bar{r}(\cdot, 0)$, is zero function and will not be learned in the second stage. We will provide strong guarantees for two-stage surrogate losses, provided that the first-stage loss function $\ell_1$ admits an $\mathcal{H}$-consistency bound, and the second-stage loss function $\ell_2$ admits an $\overline{\mathcal{R}}$-consistency bound.

**Theorem 6** ($(\mathcal{H},\mathcal{R})$-**consistency bounds for predictor-rejector two-stage surrogates**). *Assume that $\ell_1$ admits an $\mathcal{H}$-consistency bound and $\ell_2$ admits an $\overline{\mathcal{R}}$-consistency bound with respect to the multi-class zero-one classification loss $\ell_{0-1}$ respectively. Thus, there are non-decreasing concave*

*functions $\Gamma_1$ and $\Gamma_2$ such that, for all $h \in \mathcal{H}$ and $\overline{r} \in \overline{\mathcal{R}}$, we have*

$$\mathcal{E}_{\ell_{0-1}}(h) - \mathcal{E}_{\ell_{0-1}}^*(\mathcal{H}) + \mathcal{M}_{\ell_{0-1}}(\mathcal{H}) \leq \Gamma_1\big(\mathcal{E}_{\ell_1}(h) - \mathcal{E}_{\ell_1}^*(\mathcal{H}) + \mathcal{M}_{\ell_1}(\mathcal{H})\big)$$

$$\mathcal{E}_{\ell_{0-1}}(\overline{r}) - \mathcal{E}_{\ell_{0-1}}^*(\overline{\mathcal{R}}) + \mathcal{M}_{\ell_{0-1}}(\overline{\mathcal{R}}) \leq \Gamma_2\big(\mathcal{E}_{\ell_2}(\overline{r}) - \mathcal{E}_{\ell_2}^*(\overline{\mathcal{R}}) + \mathcal{M}_{\ell_2}(\overline{\mathcal{R}})\big).$$

*Then, the following holds for all $h \in \mathcal{H}$ and $r \in \mathcal{R}$:*

$$\mathcal{E}_{\mathsf{L}_{\mathrm{def}}}(h, r) - \mathcal{E}_{\mathsf{L}_{\mathrm{def}}}^*(\mathcal{H}, \mathcal{R}) + \mathcal{M}_{\mathsf{L}_{\mathrm{def}}}(\mathcal{H}, \mathcal{R})$$

$$\leq \Gamma_1\big(\mathcal{E}_{\ell_1}(h) - \mathcal{E}_{\ell_1}^*(\mathcal{H}) + \mathcal{M}_{\ell_1}(\mathcal{H})\big) + \left(1 + \sum_{j=1}^{n_e} \overline{c}_j\right)\Gamma_2\left(\frac{\mathcal{E}_{\mathsf{L}_h}(r) - \mathcal{E}_{\mathsf{L}_h}^*(\mathcal{R}) + \mathcal{M}_{\mathsf{L}_h}(\mathcal{R})}{\sum_{j=1}^{n_e} \underline{c}_j}\right),$$

*where the constant factors $\left(1 + \sum_{j=1}^{n_e} \overline{c}_j\right)$ and $\frac{1}{\sum_{j=1}^{n_e} \underline{c}_j}$ can be removed when $\Gamma_2$ is linear.*

As with the score-based setting, a specific instance of Theorem 6 holds for the case where $\mathcal{E}_{\ell_1}^*(\mathcal{H}) = \mathcal{E}_{\ell_1}^*(\mathcal{H}_{\mathrm{all}})$ and $\mathcal{E}_{\mathsf{L}_h}^*(\mathcal{R}) = \mathcal{E}_{\mathsf{L}_h}^*(\mathcal{R}_{\mathrm{all}})$, ensuring that the Bayes-error coincides with the best-in-class error and, consequently, $\mathcal{M}_{\ell_1}(\mathcal{H}) = \mathcal{M}_{\mathsf{L}_h}(\mathcal{R}) = 0$. In these cases, when the estimation error of the first-stage surrogate loss, $\mathcal{E}_{\ell_1}(h) - \mathcal{E}_{\ell_1}^*(\mathcal{H})$, is reduced to $\epsilon_1$, and the estimation error of the second-stage surrogate loss, $\mathcal{E}_{\mathsf{L}_h}(r) - \mathcal{E}_{\mathsf{L}_h}^*(\mathcal{R})$, is reduced to $\epsilon_2$, the estimation error of the deferral loss, $\mathcal{E}_{\mathsf{L}_{\mathrm{def}}}(h, r) - \mathcal{E}_{\mathsf{L}_{\mathrm{def}}}^*(\mathcal{H}, \mathcal{R})$, is upper bounded by

$$\Gamma_1(\epsilon_1) + \left(1 + \sum_{j=1}^{n_e} \overline{c}_j\right)\Gamma_2\left(\frac{\epsilon_2}{\sum_{j=1}^{n_e} \underline{c}_j}\right).$$

Next, we show that our two-stage surrogate losses are realizable $(\mathcal{H}, \mathcal{R})$-consistent. We say that the distribution is $(\mathcal{H}, \mathcal{R})$-*realizable*, if there exists a zero error solution $(h^*, r^*) \in \mathcal{H} \times \mathcal{R}$ with $\mathcal{E}_{\mathsf{L}_{\mathrm{def}}}(h^*, r^*) = 0$.

**Theorem 7** (**Realizable $(\mathcal{H}, \mathcal{R})$-consistency for predictor-rejector two-stage surrogates**). *Assume that $\mathcal{H}$ and $\mathcal{R}$ is closed under scaling and $c_j(x, y) = \beta_j, \forall(x, y) \in \mathcal{X} \times \mathcal{Y}$. Let $\ell_1$ and $\ell_2$ be the logistic loss. Let $\hat{h}$ be the minimizer of $\mathcal{E}_{\ell_1}$ and $\hat{r}$ be the minimizer of $\mathcal{E}_{\mathsf{L}_{\hat{h}}}$. Then, the following holds for any $(\mathcal{H}, \mathcal{R})$-realizable distribution,*

$$\mathcal{E}_{\mathsf{L}_{\mathrm{def}}}(\hat{h}, \hat{r}) = 0.$$

The proof is included in Appendix H. Theorem 7 suggests that the two-stage surrogate loss is realizable consistent: when the estimation error of the first-stage surrogate loss $\mathcal{E}_{\ell_1}(h_n) - \mathcal{E}_{\ell_1}^*(\mathcal{H}) \xrightarrow{n \to +\infty} 0$, and the estimation error of the second-stage surrogate loss $\mathcal{E}_{\mathsf{L}_h}(r_n) - \mathcal{E}_{\mathsf{L}_h}^*(\mathcal{R}) \xrightarrow{n \to +\infty} 0$, the estimation error of the deferral loss, $\mathcal{E}_{\mathsf{L}_{\mathrm{def}}}(h_n, r_n) - \mathcal{E}_{\mathsf{L}_{\mathrm{def}}}^*(\mathcal{H}, \mathcal{R}) \xrightarrow{n \to +\infty} 0$. By Theorem 6 and Theorem 7, in the predictor-rejector setting, we also effectively find both Bayes-consistent and realizable consistent surrogate losses with multiple experts when only the inference cost $(\beta_j)$ exists.

Note that while Sections 3 and 4 both propose new two-stage algorithms based on $\mathcal{H}$-consistent surrogate losses, they differ in an important way. Section 3 learns with deferral in a score-based framework, where deferral is associated with extra scores. In contrast, Section 4 learns with deferral in a predictor-rejector setting, where deferral corresponds to a separate function. These represent two distinct learning frameworks that have been studied in the literature. Deriving consistent surrogate losses in the predictor-rejector setting has historically been challenging for traditional single-stage scenarios, leading many to opt for the score-based approach.

We should also highlight that our $\mathcal{H}$-consistency bounds in Theorems 1 and 6 can be used to derive finite sample estimation bounds for the minimizer of the surrogate loss over a hypothesis set $\mathcal{H}$. This is achieved by upper bounding the estimation error of the minimizer of the surrogate loss using standard Rademacher complexity bounds (see [Mao et al., 2023h]).

## 5   Experiments

In this section, we report the results of our experiments on CIFAR-10 [Krizhevsky, 2009] and SVHN [Netzer et al., 2011] datasets to test the effectiveness of our proposed algorithms for two-stage

Table 4: Accuracy of deferral with multiple experts: mean ± standard deviation over three runs.

| Dataset | Base cost | Base model | Single expert | Two experts | Three experts |
|---------|-----------|------------|---------------|-------------|---------------|
| SVHN | ✗ | 91.12 | 91.85 ± 0.01% | 92.77 ± 0.02% | 93.30 ± 0.02% |
| CIFAR-10 | ✗ | 70.56 | 72.63 ± 0.20% | 75.84 ± 0.35% | 77.68 ± 0.07% |
| SVHN | ✓ | 91.12 | 91.66 ± 0.01% | 92.05 ± 0.10% | 92.19 ± 0.03% |
| CIFAR-10 | ✓ | 70.56 | 71.73 ± 0.06% | 72.31 ± 0.31% | 72.42 ± 0.12% |

learning to defer with multiple experts. We evaluated the overall accuracy of the learned pairs of predictor and deferral model across different scenarios involving varying the number of experts, where the predictor is pre-learned in the first stage and the deferral is subsequently learned using our proposed surrogate loss. We find that as the number of experts increases, the overall accuracy of the learned pairs also increases, in both scenarios with zero and non-zero base costs. This observation highlights the significance of using a multiple expert framework in our approach and the effectiveness of our surrogate loss within the framework.

We used ResNet architectures [He et al., 2016] for the prediction model, the deferral model and expert models. More precisely, we used ResNet-4 for both the predictor and the deferral. We adopted three expert models: ResNet-10, ResNet-16, ResNet-28 with increasing capacity. For training, we used the Adam optimizer [Kingma and Ba, 2014] with a batch size of 128 and weight decay $1 \times 10^{-4}$. Training was run for 15 epochs for SVHN and 50 epochs for CIFAR-10 with the default learning rate. No data augmentation was used in our experiments. We used our two-stage surrogate loss (3) with the logistic loss $\ell = \ell_{\log}$ to train the deferral model ResNet-4, with a pre-learned predictor ResNet-4 trained using logistic loss. A check mark indicates the presence of a base cost in the cost function, whereas a cross mark signifies its absence. We first set the cost function to be $\mathbb{1}_{h_j(x) \neq y}$ without a base cost. Next, for the experimental results shown in the last two row of Table 4, we chose base costs $\beta_j$ associated with each expert model as: 0.1, 0.12, 0.14 increasing with model capacity for SVHN and 0.3, 0.32, 0.34 increasing with model capacity for CIFAR-10. A base cost value that is close to the misclassification loss can strike a balance between improving accuracy and maintaining the ratio of deferral. We observed that other neighboring values lead to similar results. Note that the accuracy here refers to the overall accuracy of the learned pairs of predictor and deferral model. It is related to the deferral loss. Specifically, in the absence of the base cost, the accuracy aligns precisely with one minus the expected deferral loss. The results of Table 4 demonstrate the effectiveness of our proposed algorithms for two-stage learning to defer with multiple experts.

To the best of our knowledge, our study pioneers the exploration of a two-stage learning approach for deferral, a framework that is essential in numerous practical applications. Thus, we are unaware of any established baselines within this context.

It is important to underscore the differences between our learning scenario and those presented in [Okati et al., 2021, Narasimhan et al., 2022]. While both of them involve two phases, their methodologies are considerably different from ours. Okati et al. [2021] required conditional probabilities paired with loss estimates from the expert—a component not available in our framework, as emphasized by Mozannar et al. [2023]. On the other hand, Narasimhan et al. [2022] proposed a post-hoc correction for single-stage learning to defer surrogate losses. This approach, however, is not applicable to a pre-trained predictor from the standard multi-class classification. In contrast, our work focuses on enhancing the pre-trained predictor within the standard framework.

A limitation of our study is that the cost function used within the deferral loss is not fixed, and is typically determined through cross-validation in practice. There exists potential to introduce a principled method for selecting the cost function, which we have reserved for future research.

## 6 Conclusion

We introduced a novel family of surrogate loss functions and algorithms for a crucial two-stage learning to defer approach with multiple experts. We proved that these surrogate losses are supported by $\mathcal{H}$-consistency bounds and established their realizable $\mathcal{H}$-consistency properties for a constant cost function. This work paves the way for comparing different surrogate losses and cost functions within our framework. Further exploration, both theoretically and empirically, holds the potential to identify optimal choices for these quantities across diverse tasks.

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

# Contents of Appendix

# A    Related work

The scenario of *single-stage learning to defer* has been extensively explored in previous research. The initial studies focused on the problem of abstention and introduced various approaches such as *confidence-based abstention* [Chow, 1957, 1970, Herbei and Wegkamp, 2005, Bartlett and Wegkamp, 2008, Grandvalet et al., 2008, Yuan and Wegkamp, 2010, 2011, Ramaswamy et al., 2018, Ni et al., 2019], *selective classification* [El-Yaniv et al., 2010, Wiener and El-Yaniv, 2011, Kalai et al., 2012, Geifman and El-Yaniv, 2017, 2019, Ziyin et al., 2019, Acar et al., 2020, Gangrade et al., 2021], a *predictor-rejector* framework for abstention [Cortes et al., 2016a,b, Charoenphakdee et al., 2021, Cortes et al., 2023, Mohri et al., 2023, Mao et al., 2023c], and a *score-based setting* for abstention [Mozannar and Sontag, 2020, Raman and Yee, 2021, Liu et al., 2022, Verma and Nalisnick, 2022, Charusaie et al., 2022, Cao et al., 2022, Mao et al., 2023f, Verma et al., 2023, Mao et al., 2023b, Mozannar et al., 2023].

Another line of research is centered around the joint learning of prediction and deferral functions. Several publications by Madras et al. [2018], Raghu et al. [2019], Wilder et al. [2021], Pradier et al. [2021], Keswani et al. [2021] delve into this topic, considering single-stage learning to defer and its variants. Additionally, the concept of learning to defer has been explored in different scenarios, including combining human and machine predictions, investigating human preferences, regression problems, reinforcement learning, and more [Kamar et al., 2012, Tan et al., 2018, Kleinberg et al., 2018, Bansal et al., 2021, De et al., 2020, Straitouri et al., 2021, Zhao et al., 2021, Joshi et al., 2021, Gao et al., 2021, Mozannar et al., 2022, Hemmer et al., 2023, Narasimhan et al., 2023]. However, in practice, a predictor such as an LLM is already available and retraining one in conjunction with a deferral function could be prohibitively costly: depending on its size and the amount of data used, retraining could take several weeks or months. Thus, the single-stage learning to defer scenario and its solutions often do not align with the practical challenges encountered in real-world applications.

Alternative post-hoc methods have been proposed to address the learning to defer problem. Okati et al. [2021] proposed an iterative approach optimizing a predictor and a rejector over multiple epochs. Within each epoch, first the predictor is trained on points where its loss is lower than that of a human expert; second, the rejector is fitted to predict which of the predictor or the human expert has a lower loss. Narasimhan et al. [2022] suggested a post-hoc correction to the single-stage learning to defer surrogate losses, specifically the cost-sensitive softmax cross-entropy (CSS) surrogate loss in [Mozannar and Sontag, 2020] and the one-versus-all (OvA) surrogate loss in [Verma and Nalisnick, 2022] for cases where they suffer from underfitting. However, as with the single-stage learning to defer solutions, post-hoc approaches do not apply to scenarios where an existing predictor, pre-trained using a standard classification loss function such as cross-entropy, is already available.

A key criterion for surrogate losses in the scenario of learning to defer is Bayes-consistency (also known as consistency) [Zhang, 2004, Bartlett et al., 2006, Steinwart, 2007, Mohri et al., 2018]. This property guarantees that minimizing the surrogate loss over the family of measurable functions leads to the minimization of the deferral loss. The surrogate losses proposed in [Mozannar and Sontag, 2020, Verma and Nalisnick, 2022] have shown to be Bayes-consistent for deferral. However, Bayes-consistency is a property associated with the family of all measurable functions, which of course is considerably broader than the hypothesis sets typically used in learning algorithms, including linear hypothesis sets and the family of neural networks.

Instead, Long and Servedio [2013], Kuznetsov et al. [2014], Zhang and Agarwal [2020] proposed a notion of realizable $\mathcal{H}$-consistency, that is consistency associated with a specific hypothesis set in the realizable scenario. Mozannar et al. [2023] recently showed that existing Bayes-consistent surrogate losses in [Mozannar and Sontag, 2020, Verma and Nalisnick, 2022] are not realizable $\mathcal{H}$-consistent for learning with deferral, which can pose significant challenges when learning with a restricted hypothesis set $\mathcal{H}$, even for simple linear models. Instead, they proposed a new surrogate loss that is realizable $\mathcal{H}$-consistent when $\mathcal{H}$ is closed under scaling. However, they also observed that the loss function of Madras et al. [2018], which is not Bayes-consistent, is actually realizable $\mathcal{H}$-consistent. They acknowledged their inability to prove or disprove whether their proposed surrogate loss is Bayes-consistent. Consequently, it has remained an open problem to identify a surrogate loss that is both consistent and realizable-consistent.

In recent work, Verma et al. [2023] proposed the first Bayes-consistent surrogate losses in the scenario of learning to defer with *multiple experts* [Hemmer et al., 2022, Keswani et al., 2021, Kerrigan et al.,

2021, Straitouri et al., 2022, Benz and Rodriguez, 2022]. This scenario is more attractive and significant in applications such as large language models, where multiple models are often available for deferral. However, the surrogate losses proposed by the authors do not benefit from realizable $\mathcal{H}$-consistency, even in the single-expert setting, since they are a straightforward generalization of those of Mozannar and Sontag [2020] and Verma and Nalisnick [2022].

In summary, the problem of learning to defer in a single-stage scenario has been extensively studied, but it is often impractical in real-world applications. Post-hoc methods and surrogate losses have been explored, but the challenge remains to find a surrogate loss that is both consistent and realizable-consistent. Recent research has made progress in the scenario of learning to defer with multiple experts but has not achieved realizable $\mathcal{H}$-consistency even in a single-expert setting.

# B    Examples of two-stage score-based surrogate losses

**Example:** $\ell_2 = \ell_{\exp}$. For $\ell_2(\overline{h}_d, x, y) = \ell_{\exp}(\overline{h}_d, x, y) = \sum_{y' \neq y} e^{\overline{h}_d(x, y') - \overline{h}_d(x, y)}$, by (3), we have

$$\mathsf{L}_{h_p}(h_d, x, y)$$

$$= \mathbb{1}_{\mathsf{h}_p(x) = y} \ell_2(\overline{h}_d, x, 0) + \sum_{j=1}^{n_e} \overline{c}_j(x, y) \ell_2(\overline{h}_d, x, j)$$

$$= \mathbb{1}_{\mathsf{h}_p(x) = y} \sum_{y' \neq 0} e^{\overline{h}_d(x, y') - \overline{h}_d(x, 0)} + \sum_{j=1}^{n_e} \overline{c}_j(x, y) \sum_{y' \neq j} e^{\overline{h}_d(x, y') - \overline{h}_d(x, j)}$$

$$= \mathbb{1}_{\mathsf{h}_p(x) = y} \sum_{i=1}^{n_e} e^{h(x, n+i) - \max_{y \in \mathcal{Y}} h(x, y)} + \sum_{j=1}^{n_e} \overline{c}_j(x, y) \left[ \sum_{i=1, i \neq j}^{n_e} e^{h(x, n+i) - h(x, n+j)} + e^{\max_{y \in \mathcal{Y}} h(x, y) - h(x, n+j)} \right].$$

**Example:** $\ell_2 = \ell_{\log}$. For $\ell_2(\overline{h}_d, x, y) = \ell_{\log}(\overline{h}_d, x, y) = \log\left( \sum_{y' \in \mathcal{Y} \cup \{0\}} e^{\overline{h}_d(x, y') - \overline{h}_d(x, y)} \right)$, by (3), we have

$$\mathsf{L}_{h_p}(h_d, x, y)$$

$$= \mathbb{1}_{\mathsf{h}_p(x) = y} \ell_2(\overline{h}_d, x, 0) + \sum_{j=1}^{n_e} \overline{c}_j(x, y) \ell_2(\overline{h}_d, x, j)$$

$$= \mathbb{1}_{\mathsf{h}_p(x) = y} \log\left( \sum_{y' \in \mathcal{Y} \cup \{0\}} e^{\overline{h}_d(x, y') - \overline{h}_d(x, 0)} \right) + \sum_{j=1}^{n_e} \overline{c}_j(x, y) \log\left( \sum_{y' \in \mathcal{Y} \cup \{0\}} e^{\overline{h}_d(x, y') - \overline{h}_d(x, j)} \right)$$

$$= -\mathbb{1}_{\mathsf{h}_p(x) = y} \log\left( \frac{e^{\max_{y \in \mathcal{Y}} h(x, y)}}{e^{\max_{y \in \mathcal{Y}} h(x, y)} + \sum_{i=1}^{n_e} e^{h(x, n+i)}} \right) - \sum_{j=1}^{n_e} \overline{c}_j(x, y) \log\left( \frac{e^{h(x, n+j)}}{e^{\max_{y \in \mathcal{Y}} h(x, y)} + \sum_{i=1}^{n_e} e^{h(x, n+i)}} \right).$$

**Example:** $\ell_2 = \ell_{\text{gce}}$. For $\ell_2(\overline{h}_d, x, y) = \ell_{\text{gce}}(\overline{h}_d, x, y) = \frac{1}{\alpha}\left[ 1 - \left[ \frac{e^{\overline{h}_d(x, y)}}{\sum_{y' \in \mathcal{Y} \cup \{0\}} e^{\overline{h}_d(x, y')}} \right]^{\alpha} \right], \alpha \in (0, 1)$, by (3), we have

$$\mathsf{L}_{h_p}(h_d, x, y)$$

$$= \mathbb{1}_{\mathsf{h}_p(x) = y} \ell_2(\overline{h}_d, x, 0) + \sum_{j=1}^{n_e} \overline{c}_j(x, y) \ell_2(\overline{h}_d, x, j)$$

$$= \mathbb{1}_{\mathsf{h}_p(x) = y} \frac{1}{\alpha}\left[ 1 - \left[ \frac{e^{\overline{h}_d(x, 0)}}{\sum_{y' \in \mathcal{Y} \cup \{0\}} e^{\overline{h}_d(x, y')}} \right]^{\alpha} \right] + \sum_{j=1}^{n_e} \overline{c}_j(x, y) \frac{1}{\alpha}\left[ 1 - \left[ \frac{e^{\overline{h}_d(x, j)}}{\sum_{y' \in \mathcal{Y} \cup \{0\}} e^{\overline{h}_d(x, y')}} \right]^{\alpha} \right]$$

$$= \mathbb{1}_{\mathsf{h}_p(x) = y} \frac{1}{\alpha}\left[ 1 - \left[ \frac{e^{\max_{y \in \mathcal{Y}} h(x, y)}}{e^{\max_{y \in \mathcal{Y}} h(x, y)} + \sum_{i=1}^{n_e} e^{h(x, n+i)}} \right]^{\alpha} \right] + \sum_{j=1}^{n_e} \overline{c}_j(x, y) \frac{1}{\alpha}\left[ 1 - \left[ \frac{e^{h(x, n+j)}}{e^{\max_{y \in \mathcal{Y}} h(x, y)} + \sum_{i=1}^{n_e} e^{h(x, n+i)}} \right]^{\alpha} \right].$$

**Example:** $\ell_2 = \ell_{\mathrm{mae}}$. For $\ell_2(\overline{h}_d, x, y) = \ell_{\mathrm{mae}}(\overline{h}_d, x, y) = 1 - \frac{e^{\overline{h}_d(x,y)}}{\sum_{y' \in \mathcal{Y} \cup \{0\}} e^{\overline{h}_d(x,y')}}$, by (3), we have

$$
\mathsf{L}_{h_p}(h_d, x, y)
$$

$$
= \mathbb{1}_{\mathsf{h}_p(x)=y}\, \ell_2(\overline{h}_d, x, 0) + \sum_{j=1}^{n_e} \bar{c}_j(x, y) \ell_2(\overline{h}_d, x, j)
$$

$$
= \mathbb{1}_{\mathsf{h}_p(x)=y} \left( 1 - \frac{e^{\overline{h}_d(x,0)}}{\sum_{y' \in \mathcal{Y} \cup \{0\}} e^{\overline{h}_d(x,y')}} \right) + \sum_{j=1}^{n_e} \bar{c}_j(x, y) \left( 1 - \frac{e^{\overline{h}_d(x,j)}}{\sum_{y' \in \mathcal{Y} \cup \{0\}} e^{\overline{h}_d(x,y')}} \right)
$$

$$
= \mathbb{1}_{\mathsf{h}_p(x)=y} \left[ 1 - \frac{e^{\max_{y \in \mathcal{Y}} h(x,y)}}{e^{\max_{y \in \mathcal{Y}} h(x,y)} + \sum_{i=1}^{n_e} e^{h(x,n+i)}} \right] + \sum_{j=1}^{n_e} \bar{c}_j(x, y) \left[ 1 - \frac{e^{h(x,n+j)}}{e^{\max_{y \in \mathcal{Y}} h(x,y)} + \sum_{i=1}^{n_e} e^{h(x,n+i)}} \right].
$$

## C Examples of two-stage predictor-rejector surrogate losses

**Example:** $\ell_2 = \ell_{\exp}$. For $\ell_2(\overline{r}, x, y) = \ell_{\exp}(\overline{r}, x, y) = \sum_{y' \neq y} e^{\overline{r}(x,y') - \overline{r}(x,y)}$, by (5), we have

$$
\mathsf{L}_h(r, x, y)
$$

$$
= \mathbb{1}_{\mathsf{h}(x)=y}\, \ell_2(\overline{r}, x, 0) + \sum_{j=1}^{n_e} \bar{c}_j(x, y) \ell_2(\overline{r}, x, j)
$$

$$
= \mathbb{1}_{\mathsf{h}(x)=y} \sum_{y' \neq 0} e^{\overline{r}(x,y') - \overline{r}(x,0)} + \sum_{j=1}^{n_e} \bar{c}_j(x, y) \sum_{y' \neq j} e^{\overline{r}(x,y') - \overline{r}(x,j)}
$$

$$
= \mathbb{1}_{\mathsf{h}(x)=y} \sum_{i=1}^{n_e} e^{-r_i(x)} + \sum_{j=1}^{n_e} \bar{c}_j(x, y) \left[ \sum_{i=1, i \neq j}^{n_e} e^{r_j(x) - r_i(x)} + e^{r_j(x)} \right].
$$

**Example:** $\ell_2 = \ell_{\log}$. For $\ell_2(\overline{r}, x, y) = \ell_{\log}(\overline{r}, x, y) = \log\left( \sum_{y' \in \mathcal{Y} \cup \{0\}} e^{\overline{r}(x,y') - \overline{r}(x,y)} \right)$, by (5), we have

$$
\mathsf{L}_h(r, x, y)
$$

$$
= \mathbb{1}_{\mathsf{h}(x)=y}\, \ell_2(\overline{r}, x, 0) + \sum_{j=1}^{n_e} \bar{c}_j(x, y) \ell_2(\overline{r}, x, j)
$$

$$
= \mathbb{1}_{\mathsf{h}(x)=y} \log\left( \sum_{y' \in \mathcal{Y} \cup \{0\}} e^{\overline{r}(x,y') - \overline{r}(x,0)} \right) + \sum_{j=1}^{n_e} \bar{c}_j(x, y) \log\left( \sum_{y' \in \mathcal{Y} \cup \{0\}} e^{\overline{r}(x,y') - \overline{r}(x,j)} \right)
$$

$$
= -\mathbb{1}_{\mathsf{h}(x)=y} \log\left( \frac{1}{1 + \sum_{i=1}^{n_e} e^{-r_i(x)}} \right) - \sum_{j=1}^{n_e} \bar{c}_j(x, y) \log\left( \frac{e^{-r_j(x)}}{1 + \sum_{i=1}^{n_e} e^{-r_i(x)}} \right).
$$

**Example:** $\ell_2 = \ell_{\mathrm{gce}}$. For $\ell_2(\overline{r}, x, y) = \ell_{\mathrm{gce}}(\overline{r}, x, y) = \frac{1}{\alpha}\left[ 1 - \left[ \frac{e^{\overline{r}(x,y)}}{\sum_{y' \in \mathcal{Y} \cup \{0\}} e^{\overline{r}(x,y')}} \right]^{\alpha} \right], \alpha \in (0,1)$, by (5), we have

$$
\mathsf{L}_h(r, x, y)
$$

$$
= \mathbb{1}_{\mathsf{h}(x)=y}\, \ell_2(\overline{r}, x, 0) + \sum_{j=1}^{n_e} \bar{c}_j(x, y) \ell_2(\overline{r}, x, j)
$$

$$
= \mathbb{1}_{\mathsf{h}(x)=y} \frac{1}{\alpha}\left[ 1 - \left[ \frac{e^{\overline{r}(x,0)}}{\sum_{y' \in \mathcal{Y} \cup \{0\}} e^{\overline{r}(x,y')}} \right]^{\alpha} \right] + \sum_{j=1}^{n_e} \bar{c}_j(x, y) \frac{1}{\alpha}\left[ 1 - \left[ \frac{e^{\overline{r}(x,j)}}{\sum_{y' \in \mathcal{Y} \cup \{0\}} e^{\overline{r}(x,y')}} \right]^{\alpha} \right]
$$

$$
= \mathbb{1}_{\mathsf{h}(x)=y} \frac{1}{\alpha}\left[ 1 - \left[ \frac{1}{1 + \sum_{i=1}^{n_e} e^{-r_i(x)}} \right]^{\alpha} \right] + \sum_{j=1}^{n_e} \bar{c}_j(x, y) \frac{1}{\alpha}\left[ 1 - \left[ \frac{e^{-r_j(x)}}{1 + \sum_{i=1}^{n_e} e^{-r_i(x)}} \right]^{\alpha} \right].
$$

**Example:** $\ell_2 = \ell_{\mathrm{mae}}$. For $\ell_2(\overline{r}, x, y) = \ell_{\mathrm{mae}}(\overline{r}, x, y) = 1 - \frac{e^{\overline{r}(x,y)}}{\sum_{y' \in \mathcal{Y} \cup \{0\}} e^{\overline{r}(x,y')}}$, by (5), we have

$$
\begin{aligned}
&\mathsf{L}_h(r, x, y) \\
&= \mathbb{1}_{\mathsf{h}(x)=y}\, \ell_2(\overline{r}, x, 0) + \sum_{j=1}^{n_e} \overline{c}_j(x, y)\ell_2(\overline{r}, x, j) \\
&= \mathbb{1}_{\mathsf{h}(x)=y}\left(1 - \frac{e^{\overline{r}(x,0)}}{\sum_{y' \in \mathcal{Y} \cup \{0\}} e^{\overline{r}(x,y')}}\right) + \sum_{j=1}^{n_e} \overline{c}_j(x, y)\left(1 - \frac{e^{\overline{r}(x,j)}}{\sum_{y' \in \mathcal{Y} \cup \{0\}} e^{\overline{r}(x,y')}}\right) \\
&= \mathbb{1}_{\mathsf{h}(x)=y}\left[1 - \frac{1}{1 + \sum_{i=1}^{n_e} e^{-r_i(x)}}\right] + \sum_{j=1}^{n_e} \overline{c}_j(x, y)\left[1 - \frac{e^{-r_j(x)}}{1 + \sum_{i=1}^{n_e} e^{-r_i(x)}}\right].
\end{aligned}
$$

## D  Proof of $\mathcal{H}$-consistency bounds for score-based two-stage surrogate losses (Theorem 1)

**Theorem 1** ($\mathcal{H}$-consistency bounds for score-based two-stage surrogates). *Assume that $\ell_1$ admits an $\mathcal{H}_p$-consistency bound and $\ell_2$ admits an $\overline{\mathcal{H}}_d$-consistency bound with respect to the multi-class zero-one classification loss $\ell_{0-1}$ respectively. Thus, there are non-decreasing concave functions $\Gamma_1$ and $\Gamma_2$ such that, for all $h_p \in \mathcal{H}_p$ and $\overline{h}_d \in \overline{\mathcal{H}}_d$, we have*

$$
\begin{aligned}
\mathcal{E}_{\ell_{0-1}}(h_p) - \mathcal{E}^*_{\ell_{0-1}}(\mathcal{H}_p) + \mathcal{M}_{\ell_{0-1}}(\mathcal{H}_p) &\leq \Gamma_1\big(\mathcal{E}_{\ell_1}(h_p) - \mathcal{E}^*_{\ell_1}(\mathcal{H}_p) + \mathcal{M}_{\ell_1}(\mathcal{H}_p)\big) \\
\mathcal{E}_{\ell_{0-1}}(\overline{h}_d) - \mathcal{E}^*_{\ell_{0-1}}(\overline{\mathcal{H}}_d) + \mathcal{M}_{\ell_{0-1}}(\overline{\mathcal{H}}_d) &\leq \Gamma_2\big(\mathcal{E}_{\ell_2}(\overline{h}_d) - \mathcal{E}^*_{\ell_2}(\overline{\mathcal{H}}_d) + \mathcal{M}_{\ell_2}(\overline{\mathcal{H}}_d)\big).
\end{aligned}
$$

*Then, the following holds for all $h \in \mathcal{H}$:*

$$
\begin{aligned}
&\mathcal{E}_{\mathsf{L}_{\mathrm{def}}}(h) - \mathcal{E}^*_{\mathsf{L}_{\mathrm{def}}}(\mathcal{H}) + \mathcal{M}_{\mathsf{L}_{\mathrm{def}}}(\mathcal{H}) \\
&\leq \Gamma_1\big(\mathcal{E}_{\ell_1}(h_p) - \mathcal{E}^*_{\ell_1}(\mathcal{H}_p) + \mathcal{M}_{\ell_1}(\mathcal{H}_p)\big) + \left(1 + \sum_{j=1}^{n_e} \overline{c}_j\right)\Gamma_2\left(\frac{\mathcal{E}_{\mathsf{L}_{h_p}}(h_d) - \mathcal{E}^*_{\mathsf{L}_{h_p}}(\mathcal{H}_d) + \mathcal{M}_{\mathsf{L}_{h_p}}(\mathcal{H}_d)}{\sum_{j=1}^{n_e} \underline{c}_j}\right).
\end{aligned}
$$

*Furthermore, constant factors $\left(1 + \sum_{j=1}^{n_e} \overline{c}_j\right)$ and $\frac{1}{\sum_{j=1}^{n_e} \underline{c}_j}$ can be removed when $\Gamma_2$ is linear.*

*Proof.* If $\mathsf{h}(x) \in [n]$, then $\mathsf{h}(x) = \mathsf{h}_\mathsf{p}(x)$. Thus, the learning to defer loss can be expressed as follows:

$$
\begin{aligned}
\mathsf{L}_{\mathrm{def}}(h, x, y) &= \mathbb{1}_{\mathsf{h}(x) \neq y}\mathbb{1}_{\mathsf{h}(x) \in [n]} + \sum_{j=1}^{n_e} c_j(x, y)\mathbb{1}_{\mathsf{h}(x) = n+j} \\
&= \mathbb{1}_{\mathsf{h}_\mathsf{p}(x) \neq y}\mathbb{1}_{\mathsf{h}(x) \in [n]} + \sum_{j=1}^{n_e} c_j(x, y)\mathbb{1}_{\mathsf{h}(x) = n+j}.
\end{aligned}
$$

Let $\overline{c}_0(x, y) = \mathbb{1}_{\mathsf{h}_\mathsf{p}(x)=y}$. Since $h = (h_p, h_d)$, we can rewrite $\mathcal{E}_{\mathsf{L}_{\mathrm{def}}}(h) - \mathcal{E}^*_{\mathsf{L}_{\mathrm{def}}}(\mathcal{H}) + \mathcal{M}_{\mathsf{L}_{\mathrm{def}}}(\mathcal{H})$ as

$$
\begin{aligned}
&\mathcal{E}_{\mathsf{L}_{\mathrm{def}}}(h) - \mathcal{E}^*_{\mathsf{L}_{\mathrm{def}}}(\mathcal{H}) + \mathcal{M}_{\mathsf{L}_{\mathrm{def}}}(\mathcal{H}) \\
&= \mathbb{E}_X\big[\mathcal{C}_{\mathsf{L}_{\mathrm{def}}}(h, x) - \mathcal{C}^*_{\mathsf{L}_{\mathrm{def}}}(\mathcal{H}, x)\big] \\
&= \mathbb{E}_X\Big[\mathcal{C}_{\mathsf{L}_{\mathrm{def}}}(h, x) - \inf_{h_d \in \mathcal{H}_d} \mathcal{C}_{\mathsf{L}_{\mathrm{def}}}(h, x) + \inf_{h_d \in \mathcal{H}_d} \mathcal{C}_{\mathsf{L}_{\mathrm{def}}}(h, x) - \mathcal{C}^*_{\mathsf{L}_{\mathrm{def}}}(\mathcal{H}, x)\Big] \quad (6)\\
&= \mathbb{E}_X\Big[\mathcal{C}_{\mathsf{L}_{\mathrm{def}}}(h, x) - \inf_{h_d \in \mathcal{H}_d} \mathcal{C}_{\mathsf{L}_{\mathrm{def}}}(h, x)\Big] + \mathbb{E}_X\Big[\inf_{h_d \in \mathcal{H}_d} \mathcal{C}_{\mathsf{L}_{\mathrm{def}}}(h, x) - \mathcal{C}^*_{\mathsf{L}_{\mathrm{def}}}(\mathcal{H}, x)\Big].
\end{aligned}
$$

Let $\overline{p}(x, j) = \frac{\mathbb{E}_y[\overline{c}_j(x,y)]}{\mathbb{E}_y\left[\sum_{j=0}^{n_e} \overline{c}_j(x,y)\right]}$ for any $j \in \{0, \ldots, n_e\}$. Note that $\overline{p}(x, \cdot)$ is the probability vector on the label space $\{0, \ldots, n_e\}$. For any $h \in \mathcal{H}$, we define $\overline{h}$ as its augmented hypothesis: $\overline{h}(x, 0) =$

$\max_{y \in \mathcal{Y}} h(x, y), \overline{h}(x, 1) = h(x, 1), \ldots, \overline{h}(x, n_e) = h(x, n_e)$. By the assumptions, we have

$$\mathcal{C}_{\mathsf{L}_{\mathrm{def}}}(h, x) - \inf_{h_d \in \mathcal{H}_d} \mathcal{C}_{\mathsf{L}_{\mathrm{def}}}(h, x)$$

$$= \mathbb{E}_y \left[ \mathbb{1}_{\mathsf{h}_{\mathsf{p}}(x) \neq y} \mathbb{1}_{\mathsf{h}(x) \in [n]} + \sum_{j=1}^{n_e} c_j(x, y) \mathbb{1}_{\mathsf{h}(x) = n+j} \right] - \inf_{h_d \in \mathcal{H}_d} \mathbb{E}_y \left[ \mathbb{1}_{\mathsf{h}_{\mathsf{p}}(x) \neq y} \mathbb{1}_{\mathsf{h}(x) \in [n]} + \sum_{j=1}^{n_e} c_j(x, y) \mathbb{1}_{\mathsf{h}(x) = n+j} \right]$$

$$= \mathbb{E}_y \left[ \sum_{j=0}^{n_e} \overline{c}_j(x, y) \right] \times \left[ \sum_{j=0}^{n_e} \overline{p}(x, j) \ell_{0-1}(\overline{h}, x, j) - \inf_{h_d \in \mathcal{H}_d} \sum_{j=0}^{n_e} \overline{p}(x, j) \ell_{0-1}(\overline{h}, x, j) \right]$$

$$\leq \mathbb{E}_y \left[ \sum_{j=0}^{n_e} \overline{c}_j(x, y) \right] \times \Gamma_2 \left[ \sum_{j=0}^{n_e} \overline{p}(x, j) \ell_2(\overline{h}, x, j) - \inf_{h_d \in \mathcal{H}_d} \sum_{j=0}^{n_e} \overline{p}(x, j) \ell_2(\overline{h}, x, j) \right]$$

(By $\overline{\mathcal{H}}_d$-consistency bounds of $\ell_2$ under assumption, $\lambda = \max_{y \in \mathcal{Y}} h(x, y)$)

$$= \mathbb{E}_y \left[ \sum_{j=0}^{n_e} \overline{c}_j(x, y) \right] \Gamma_2 \left( \frac{\mathbb{E}_y \left[ \mathsf{L}_{h_p}(h_d, x, y) \right] - \inf_{h_d \in \mathcal{H}_d} \mathbb{E}_y \left[ \mathsf{L}_{h_p}(h_d, x, y) \right]}{\mathbb{E}_y \left[ \sum_{j=0}^{n_e} \overline{c}_j(x, y) \right]} \right)$$

$$\left( \overline{p}(x, j) = \frac{\mathbb{E}_y [\overline{c}_j(x, y)]}{\mathbb{E}_y \left[ \sum_{j=0}^{n_e} \overline{c}_j(x, y) \right]}, \overline{h}(x, 0) = \max_{y \in \mathcal{Y}} h(x, y) \text{ and formulation } (3) \right)$$

$$= \mathbb{E}_y \left[ \sum_{j=0}^{n_e} \overline{c}_j(x, y) \right] \Gamma_2 \left( \frac{\mathcal{C}_{\mathsf{L}_{h_p}}(h_d, x) - \mathcal{C}^*_{\mathsf{L}_{h_p}}(\mathcal{H}_d, x)}{\mathbb{E}_y \left[ \sum_{j=0}^{n_e} \overline{c}_j(x, y) \right]} \right)$$

$$\leq \begin{cases} \Gamma_2 \left( \mathcal{C}_{\mathsf{L}_{h_p}}(h_d, x) - \mathcal{C}^*_{\mathsf{L}_{h_p}}(\mathcal{H}_d, x) \right) & \text{when } \Gamma_2 \text{ is linear} \\ \left( 1 + \sum_{j=1}^{n_e} \overline{c}_j \right) \Gamma_2 \left( \frac{\mathcal{C}_{\mathsf{L}_{h_p}}(h_d, x) - \mathcal{C}^*_{\mathsf{L}_{h_p}}(\mathcal{H}_d, x)}{\sum_{j=1}^{n_e} \underline{c}_j} \right) & \text{otherwise} \end{cases}$$

$$\left( \sum_{j=1}^{n_e} \underline{c}_j \leq \mathbb{E}_y \left[ \sum_{j=0}^{n_e} \overline{c}_j(x, y) \right] \leq 1 + \sum_{j=1}^{n_e} \overline{c}_j \text{ and } \Gamma_2 \text{ is non-decreasing} \right)$$

$$= \begin{cases} \Gamma_2 \left( \Delta \mathcal{C}_{\mathsf{L}_{h_p}, \mathcal{H}_d}(h_d, x) \right) & \text{when } \Gamma_2 \text{ is linear} \\ \left( 1 + \sum_{j=1}^{n_e} \overline{c}_j \right) \Gamma_2 \left( \frac{\Delta \mathcal{C}_{\mathsf{L}_{h_p}, \mathcal{H}_d}(h_d, x)}{\sum_{j=1}^{n_e} \underline{c}_j} \right) & \text{otherwise} \end{cases}$$

and

$$\inf_{h_d \in \mathcal{H}_d} \mathcal{C}_{\mathsf{L}_{\mathrm{def}}}(h, x) - \mathcal{C}^*_{\mathsf{L}_{\mathrm{def}}}(\mathcal{H}, x)$$

$$= \inf_{h_d \in \mathcal{H}_d} \mathcal{C}_{\mathsf{L}_{\mathrm{def}}}(h, x) - \inf_{h_p \in \mathcal{H}_p, h_d \in \mathcal{H}_d} \mathcal{C}_{\mathsf{L}_{\mathrm{def}}}(h, x)$$

$$= \inf_{h_d \in \mathcal{H}_d} \mathbb{E}_y \left[ \mathbb{1}_{\mathsf{h}_{\mathsf{p}}(x) \neq y} \mathbb{1}_{\mathsf{h}(x) \in [n]} + \sum_{j=1}^{n_e} c_j(x, y) \mathbb{1}_{\mathsf{h}(x) = n+j} \right]$$

$$\quad - \inf_{h_p \in \mathcal{H}_p, h_d \in \mathcal{H}_d} \mathbb{E}_y \left[ \mathbb{1}_{\mathsf{h}_{\mathsf{p}}(x) \neq y} \mathbb{1}_{\mathsf{h}(x) \in [n]} + \sum_{j=1}^{n_e} c_j(x, y) \mathbb{1}_{\mathsf{h}(x) = n+j} \right]$$

$$= \inf_{h_d \in \mathcal{H}_d} \mathbb{E}_y \left[ \mathbb{1}_{\mathsf{h}_{\mathsf{p}}(x) \neq y} \mathbb{1}_{\mathsf{h}(x) \in [n]} + \sum_{j=1}^{n_e} c_j(x, y) \mathbb{1}_{\mathsf{h}(x) = n+j} \right]$$

$$\quad - \inf_{h_d \in \mathcal{H}_d} \mathbb{E}_y \left[ \inf_{h_p \in \mathcal{H}_p} \mathbb{1}_{\mathsf{h}_{\mathsf{p}}(x) \neq y} \mathbb{1}_{\mathsf{h}(x) \in [n]} + \sum_{j=1}^{n_e} c_j(x, y) \mathbb{1}_{\mathsf{h}(x) = n+j} \right]$$

$$= \min \left\{ \mathbb{E}_y \left[ \mathbb{1}_{\mathsf{h}_{\mathsf{p}}(x) \neq y} \right], \min_{j \in [p]} \mathbb{E}_y [c_j(x, y)] \right\} - \min \left\{ \inf_{h_p \in \mathcal{H}_p} \mathbb{E}_y \left[ \mathbb{1}_{\mathsf{h}_{\mathsf{p}}(x) \neq y} \right], \min_{j \in [p]} \mathbb{E}_y [c_j(x, y)] \right\}$$

$$\leq \mathbb{E}_y \left[ \mathbb{1}_{\mathsf{h}_{\mathsf{p}}(x) \neq y} \right] - \inf_{h_p \in \mathcal{H}_p} \mathbb{E}_y \left[ \mathbb{1}_{\mathsf{h}_{\mathsf{p}}(x) \neq y} \right]$$

$$= \mathcal{C}_{\ell_{0-1}}(h_p, x) - \mathcal{C}^*_{\ell_{0-1}}(\mathcal{H}_p, x)$$

$$= \Delta \mathcal{C}_{\ell_{0-1}, \mathcal{H}_p}(h_p, x)$$

$$\leq \Gamma_1 \left( \Delta \mathcal{C}_{\ell_1, \mathcal{H}_p}(h_p, x) \right). \qquad \text{(By } \mathcal{H}_p\text{-consistency bounds of } \ell \text{ under assumption)}$$

Therefore, by (6), we obtain

$$
\begin{aligned}
&\mathcal{E}_{\mathsf{L}_{\mathrm{def}}}(h) - \mathcal{E}_{\mathsf{L}_{\mathrm{def}}}^*(\mathcal{H}) + \mathcal{M}_{\mathsf{L}_{\mathrm{def}}}(\mathcal{H}) \\
&\leq
\begin{cases}
\mathbb{E}_X\Big[\Gamma_2\big(\Delta\mathcal{C}_{\mathsf{L}_{h_p},\mathcal{H}_d}(h_d,x)\big)\Big] + \mathbb{E}_X\big[\Gamma_1\big(\Delta\mathcal{C}_{\ell_1,\mathcal{H}_p}(h_p,x)\big)\big] & \text{when } \Gamma_2 \text{ is linear} \\
\big(1 + \sum_{j=1}^{n_e}\overline{c}_j\big)\mathbb{E}_X\Big[\Gamma_2\Big(\frac{\Delta\mathcal{C}_{\mathsf{L}_{h_p},\mathcal{H}_d}(h_d,x)}{\sum_{j=1}^{n_e}\underline{c}_j}\Big)\Big] + \mathbb{E}_X\big[\Gamma_1\big(\Delta\mathcal{C}_{\ell_1,\mathcal{H}_p}(h_p,x)\big)\big] & \text{otherwise} \\
\end{cases} \\
&\leq
\begin{cases}
\Gamma_2\Big(\mathbb{E}_X\big[\Delta\mathcal{C}_{\mathsf{L}_{h_p},\mathcal{H}_d}(h_d,x)\big]\Big) + \Gamma_1\big(\mathbb{E}_X\big[\Delta\mathcal{C}_{\ell_1,\mathcal{H}_p}(h_p,x)\big]\big) & \text{when } \Gamma_2 \text{ is linear} \\
\big(1 + \sum_{j=1}^{n_e}\overline{c}_j\big)\Gamma_2\Big(\frac{1}{\sum_{j=1}^{n_e}\underline{c}_j}\mathbb{E}_X\big[\Delta\mathcal{C}_{\mathsf{L}_{h_p},\mathcal{H}_d}(h_d,x)\big]\Big) + \Gamma_1\big(\mathbb{E}_X\big[\Delta\mathcal{C}_{\ell_1,\mathcal{H}_p}(h_p,x)\big]\big) & \text{otherwise}
\end{cases} \\
&\hspace{10cm} (\Gamma_1 \text{ and } \Gamma_2 \text{ are concave}) \\
&=
\begin{cases}
\Gamma_1\big(\mathcal{E}_{\ell_1}(h_p) - \mathcal{E}_{\ell_1}^*(\mathcal{H}_p) + \mathcal{M}_{\ell_1}(\mathcal{H}_p)\big) + \Gamma_2\big(\mathcal{E}_{\mathsf{L}_{h_p}}(h_d) - \mathcal{E}_{\mathsf{L}_{h_p}}^*(\mathcal{H}_d) + \mathcal{M}_{\mathsf{L}_{h_p}}(\mathcal{H}_d)\big) & \text{when } \Gamma_2 \text{ is linear} \\
\Gamma_1\big(\mathcal{E}_{\ell_1}(h_p) - \mathcal{E}_{\ell_1}^*(\mathcal{H}_p) + \mathcal{M}_{\ell_1}(\mathcal{H}_p)\big) + \big(1 + \sum_{j=1}^{n_e}\overline{c}_j\big)\Gamma_2\Big(\frac{\mathcal{E}_{\mathsf{L}_{h_p}}(h_d) - \mathcal{E}_{\mathsf{L}_{h_p}}^*(\mathcal{H}_d) + \mathcal{M}_{\mathsf{L}_{h_p}}(\mathcal{H}_d)}{\sum_{j=1}^{n_e}\underline{c}_j}\Big) & \text{otherwise,}
\end{cases}
\end{aligned}
$$

which completes the proof. $\qquad\square$

## E  Proof of $\overline{\mathcal{H}}$-consistency bounds for standard surrogate loss functions (Theorem 3)

Recall that for a hypothesis $h\colon \mathcal{X} \times \mathcal{Y} \to \mathbb{R}$, we define $\overline{h}$ as its augmented hypothesis: $\overline{h}(\cdot, 0) = \lambda, \overline{h}(\cdot, 1) = h(x, 1), \ldots, \overline{h}(\cdot, n) = h(x, n)$ with some constant $\lambda \in \mathbb{R}$. We define $\overline{\mathcal{H}}$ as the hypothesis set that consists of all such augmented hypotheses of $\mathcal{H}$: $\overline{\mathcal{H}} = \big\{\overline{h} : h \in \mathcal{H}\big\}$. The prediction associated by $\overline{h} \in \overline{\mathcal{H}}$ to an input $x \in \mathcal{X}$ is denoted by $\overline{\mathsf{h}}(x)$ and defined as the element in $\mathcal{Y} \cup \{0\}$ with the highest score, $\overline{\mathsf{h}}(x) = \operatorname{argmax}_{y \in \mathcal{Y} \cup \{0\}} h(x, y)$, with an arbitrary but fixed deterministic strategy for breaking ties. For any $x \in \mathcal{X}$ and label space $\mathcal{Y} \cup \{0\}$, we will denote, by $\overline{\mathsf{H}}(x)$ the set of labels generated by hypotheses in $\overline{\mathcal{H}}$: $\overline{\mathsf{H}}(x) = \big\{\overline{\mathsf{h}}(x)\colon h \in \overline{\mathcal{H}}\big\}$. By [Awasthi et al., 2022a, Lemma 3] with label space $\mathcal{Y} \cup \{0\}$ and a conditional probability vector $p(x, \cdot)$ on $\mathcal{Y} \cup \{0\}$, the minimal conditional $\ell_{0-1}$-loss and the corresponding calibration gap can be characterized as follows.

**Lemma 8.** *For any $x \in \mathcal{X}$, the minimal conditional $\ell_{0-1}$-risk and the calibration gap for $\ell_{0-1}$ can be expressed as follows:*

$$
\mathcal{C}_{\ell_{0-1}}^*(x) = 1 - \max_{y \in \overline{\mathsf{H}}(x)} p(x, y)
$$
$$
\Delta\mathcal{C}_{\ell_{0-1}}(h, x) = \max_{y \in \overline{\mathsf{H}}(x)} p(x, y) - p(x, \mathsf{h}(x)).
$$

### E.1  Multinomial logistic loss

**Theorem 9** ($\overline{\mathcal{H}}$-consistency bound for multinomial logistic loss)**.** *Assume that $\mathcal{H}$ is symmetric and complete. Then, for any $\lambda \in \mathbb{R}$, hypothesis $\overline{h} \in \overline{\mathcal{H}}$ and any distribution,*

$$
\mathcal{E}_{\ell_{0-1}}\big(\overline{h}\big) - \mathcal{E}_{\ell_{0-1}}^*\big(\overline{\mathcal{H}}\big) \leq \sqrt{2}\Big(\mathcal{E}_{\ell_{\log}}\big(\overline{h}\big) - \mathcal{E}_{\ell_{\log}}^*\big(\overline{\mathcal{H}}\big) + \mathcal{M}_{\ell_{\log}}\big(\overline{\mathcal{H}}\big)\Big)^{\frac{1}{2}} - \mathcal{M}_{\ell_{0-1}}\big(\overline{\mathcal{H}}\big).
$$

*Proof.* For the multinomial logistic loss $\ell_{\log}$, the conditional $\ell_{\log}$-risk can be expressed as follows:

$$
\mathcal{C}_{\ell_{\log}}\big(\overline{h}, x)\big) = \sum_{y \in \mathcal{Y} \cup \{0\}} p(x, y) \log\!\left(\sum_{y' \in \mathcal{Y} \cup \{0\}} e^{\overline{h}(x, y') - \overline{h}(x, y)}\right) = - \sum_{y \in \mathcal{Y} \cup \{0\}} p(x, y) \log(\mathcal{S}(x, y))
$$

where we let $\mathcal{S}(x, y) = \frac{e^{\overline{h}(x, y)}}{\sum_{y' \in \mathcal{Y} \cup \{0\}} e^{\overline{h}(x, y')}} \in [0, 1]$ for any $y \in \mathcal{Y} \cup \{0\}$ with the constraint that $\sum_{y \in \mathcal{Y} \cup \{0\}} \mathcal{S}(x, y) = 1$. Let $y_{\max} = \operatorname{argmax}_{y \in \mathcal{Y} \cup \{0\}} p(x, y)$, where we choose the label with the same deterministic strategy for breaking ties as that of $\overline{\mathsf{h}}(x)$. For any $\overline{h} \in \mathcal{H}$ such that $\overline{\mathsf{h}}(x) \neq y_{\max}$ and $x \in \mathcal{X}$, by the symmetry and completeness of $\mathcal{H}$, we can always find a family of hypotheses

$\left\{\overline{\mathsf{h}}_\mu : \mu \in \left[-\mathcal{S}(x, y_{\max}), \mathcal{S}(x, \overline{\mathsf{h}}(x))\right]\right\} \subset \overline{\mathcal{H}}$ such that $\mathcal{S}_\mu(x, \cdot) = \frac{e^{\overline{h}_\mu(x, \cdot)}}{\sum_{y' \in \mathcal{Y} \cup \{0\}} e^{\overline{h}_\mu(x, y')}}$ take the following values:

$$\mathcal{S}_\mu(x, y) = \begin{cases} \mathcal{S}(x, y) & \text{if } y \notin \{y_{\max}, \overline{\mathsf{h}}(x)\} \\ \mathcal{S}(x, y_{\max}) + \mu & \text{if } y = \overline{\mathsf{h}}(x) \\ \mathcal{S}(x, \overline{\mathsf{h}}(x)) - \mu & \text{if } y = y_{\max}. \end{cases}$$

Note that $\mathcal{S}_\mu$ satisfies the constraint:

$$\sum_{y \in \mathcal{Y} \cup \{0\}} \mathcal{S}_\mu(x, y) = \sum_{y \in \mathcal{Y} \cup \{0\}} \mathcal{S}(x, y) = 1, \ \forall \mu \in \left[-\mathcal{S}(x, y_{\max}), \mathcal{S}(x, \overline{\mathsf{h}}(x))\right].$$

Let $\overline{h} \in \overline{\mathcal{H}}$ be a hypothesis such that $\overline{\mathsf{h}}(x) \neq y_{\max}$. By the definition and using the fact that $\overline{\mathsf{H}}(x) = \mathcal{Y} \cup \{0\}$ when $\mathcal{H}$ is symmetric, we obtain

$$\Delta\mathcal{C}_{\ell_{\log}, \overline{\mathcal{H}}}(\overline{h}, x)$$

$$= \mathcal{C}_{\ell_{\log}}(\overline{h}, x) - \mathcal{C}^*_{\ell_{\log}}(\overline{\mathcal{H}}, x)$$

$$\geq \mathcal{C}_{\ell_{\log}}(\overline{h}, x) - \inf_{\mu \in [-\mathcal{S}(x, y_{\max}), \mathcal{S}(x, \overline{\mathsf{h}}(x))]} \mathcal{C}_{\ell_{\log}}(\overline{h}_\mu, x)$$

$$= \sup_{\mu \in [-\mathcal{S}(x, y_{\max}), \mathcal{S}(x, \overline{\mathsf{h}}(x))]} \left\{ p(x, y_{\max}) \left[-\log(\mathcal{S}(x, y_{\max})) + \log(\mathcal{S}(x, \overline{\mathsf{h}}(x)) - \mu)\right] \right.$$

$$\left. + p(x, \overline{\mathsf{h}}(x)) \left[-\log(\mathcal{S}(x, \overline{\mathsf{h}}(x))) + \log(\mathcal{S}(x, y_{\max}) + \mu)\right] \right\}$$

Differentiating with respect to $\mu$ yields the optimum value $\mu^* = \frac{p(x, \overline{\mathsf{h}}(x))\mathcal{S}(x, \overline{\mathsf{h}}(x)) - p(x, y_{\max})\mathcal{S}(x, y_{\max})}{p(x, y_{\max}) + p(x, \overline{\mathsf{h}}(x))}$.
Plugging that value in the inequality gives:

$$\Delta\mathcal{C}_{\ell_{\log}, \overline{\mathcal{H}}}(\overline{h}, x) \geq p(x, y_{\max}) \log \frac{[\mathcal{S}(x, \overline{\mathsf{h}}(x)) + \mathcal{S}(x, y_{\max})]p(x, y_{\max})}{\mathcal{S}(x, y_{\max})[p(x, y_{\max}) + p(x, \overline{\mathsf{h}}(x))]}$$

$$+ p(x, \overline{\mathsf{h}}(x)) \log \frac{[\mathcal{S}(x, \overline{\mathsf{h}}(x)) + \mathcal{S}(x, y_{\max})]p(x, \overline{\mathsf{h}}(x))}{\mathcal{S}(x, \overline{\mathsf{h}}(x))[p(x, y_{\max}) + p(x, \overline{\mathsf{h}}(x))]}.$$

Differentiating with respect to $\mathcal{S}$ to show that the minimum is attained for $\mathcal{S}(x, \overline{\mathsf{h}}(x)) = \mathcal{S}(x, y_{\max})$, which implies

$$\Delta\mathcal{C}_{\ell_{\log}, \overline{\mathcal{H}}}(\overline{h}, x) \geq p(x, y_{\max}) \log \frac{2p(x, y_{\max})}{p(x, y_{\max}) + p(x, \overline{\mathsf{h}}(x))} + p(x, \overline{\mathsf{h}}(x)) \log \frac{2p(x, \overline{\mathsf{h}}(x))}{p(x, y_{\max}) + p(x, \overline{\mathsf{h}}(x))}.$$

By Pinsker's inequality, we have, for $a, b \in [0, 1]$, $a \log \frac{2a}{a+b} + b \log \frac{2b}{a+b} \geq \frac{(a-b)^2}{2(a+b)}$. Using this inequality, we obtain:

$$\Delta\mathcal{C}_{\ell_{\log}, \overline{\mathcal{H}}}(\overline{h}, x) \geq \frac{\left(p(x, \overline{\mathsf{h}}(x)) - p(x, y_{\max})\right)^2}{2\left(p(x, \overline{\mathsf{h}}(x)) + p(x, y_{\max})\right)}$$

$$\geq \frac{\left(p(x, \overline{\mathsf{h}}(x)) - p(x, y_{\max})\right)^2}{2} \qquad (0 \leq p(x, \overline{\mathsf{h}}(x)) + p(x, y_{\max}) \leq 1)$$

$$= \frac{1}{2}\left(\Delta\mathcal{C}_{\ell_{0-1}, \overline{\mathcal{H}}}(\overline{h}, x)\right)^2. \qquad (\text{by Lemma } 8 \text{ and } \overline{\mathsf{H}}(x) = \mathcal{Y} \cup \{0\})$$

Since the function $\frac{t^2}{2}$ is convex, by Jensen's inequality, we obtain for any hypothesis $\overline{h} \in \overline{\mathcal{H}}$ and any distribution,

$$\frac{\left(\mathbb{E}_X\left[\Delta\mathcal{C}_{\ell_{0-1}, \overline{\mathcal{H}}}(\overline{h}, x)\right]\right)^2}{2} \leq \mathbb{E}_X\left[\frac{\Delta\mathcal{C}_{\ell_{0-1}, \overline{\mathcal{H}}}(\overline{h}, x)^2}{2}\right] \leq \mathbb{E}_X\left[\Delta\mathcal{C}_{\ell_{\log}, \overline{\mathcal{H}}}(\overline{h}, x)\right],$$

which leads to

$$\mathcal{E}_{\ell_{0-1}}(\overline{h}) - \mathcal{E}^*_{\ell_{0-1}}(\overline{\mathcal{H}}) \leq \sqrt{2}\left(\mathcal{E}_{\ell_{\log}}(\overline{h}) - \mathcal{E}^*_{\ell_{\log}}(\overline{\mathcal{H}}) + \mathcal{M}_{\ell_{\log}}(\overline{\mathcal{H}})\right)^{\frac{1}{2}} - \mathcal{M}_{\ell_{0-1}}(\overline{\mathcal{H}}).$$

$\square$

## E.2 Sum exponential loss

**Theorem 10 ($\overline{\mathcal{H}}$-consistency bound for sum exponential loss).** *Assume that $\mathcal{H}$ is symmetric and complete. Then, for any $\lambda \in \mathbb{R}$, hypothesis $\overline{h} \in \overline{\mathcal{H}}$ and any distribution,*

$$\mathcal{E}_{\ell_{0-1}}(\overline{h}) - \mathcal{E}^*_{\ell_{0-1}}(\overline{\mathcal{H}}) \leq \sqrt{2}\big(\mathcal{E}_{\ell_{\exp}}(\overline{h}) - \mathcal{E}^*_{\ell_{\exp}}(\overline{\mathcal{H}}) + \mathcal{M}_{\ell_{\exp}}(\overline{\mathcal{H}})\big)^{\frac{1}{2}} - \mathcal{M}_{\ell_{0-1}}(\overline{\mathcal{H}}).$$

*Proof.* For the sum exponential loss $\ell_{\exp}$, the conditional $\ell_{\exp}$-risk can be expressed as follows:

$$\mathcal{C}_{\ell_{\exp}}(\overline{h}, x)) = \sum_{y \in \mathcal{Y} \cup \{0\}} p(x, y)\left(\sum_{y' \in \mathcal{Y} \cup \{0\}} e^{\overline{h}(x, y') - \overline{h}(x, y)}\right) - 1 = \sum_{y \in \mathcal{Y} \cup \{0\}} \frac{p(x, y)}{\mathcal{S}(x, y)} - 1$$

where we let $\mathcal{S}(x, y) = \frac{e^{\overline{h}(x,y)}}{\sum_{y' \in \mathcal{Y} \cup \{0\}} e^{\overline{h}(x,y')}} \in [0, 1]$ for any $y \in \mathcal{Y} \cup \{0\}$ with the constraint that $\sum_{y \in \mathcal{Y} \cup \{0\}} \mathcal{S}(x, y) = 1$. Let $y_{\max} = \mathrm{argmax}_{y \in \mathcal{Y} \cup \{0\}} p(x, y)$, where we choose the label with the highest index under the natural ordering of labels as the tie-breaking strategy. For any $\overline{h} \in \mathcal{H}$ such that $\overline{h}(x) \neq y_{\max}$ and $x \in \mathcal{X}$, by the symmetry and completeness of $\mathcal{H}$, we can always find a family of hypotheses $\{\overline{h}_\mu : \mu \in [-\mathcal{S}(x, y_{\max}), \mathcal{S}(x, \overline{h}(x))]\} \subset \overline{\mathcal{H}}$ such that $\mathcal{S}_\mu(x, \cdot) = \frac{e^{\overline{h}_\mu(x, \cdot)}}{\sum_{y' \in \mathcal{Y} \cup \{0\}} e^{\overline{h}_\mu(x, y')}}$ take the following values:

$$\mathcal{S}_\mu(x, y) = \begin{cases} \mathcal{S}(x, y) & \text{if } y \notin \{y_{\max}, \overline{h}(x)\} \\ \mathcal{S}(x, y_{\max}) + \mu & \text{if } y = \overline{h}(x) \\ \mathcal{S}(x, \overline{h}(x)) - \mu & \text{if } y = y_{\max}. \end{cases}$$

Note that $\mathcal{S}_\mu$ satisfies the constraint:

$$\sum_{y \in \mathcal{Y}} \mathcal{S}_\mu(x, y) = \sum_{y \in \mathcal{Y}} \mathcal{S}(x, y) = 1, \ \forall \mu \in [-\mathcal{S}(x, y_{\max}), \mathcal{S}(x, \overline{h}(x))].$$

Let $\overline{h} \in \overline{\mathcal{H}}$ be a hypothesis such that $\overline{h}(x) \neq y_{\max}$. By the definition and using the fact that $\overline{H}(x) = \mathcal{Y} \cup \{0\}$ when $\mathcal{H}$ is symmetric, we obtain

$$\Delta\mathcal{C}_{\ell_{\exp}, \overline{\mathcal{H}}}(\overline{h}, x)$$
$$= \mathcal{C}_{\ell_{\exp}}(\overline{h}, x) - \mathcal{C}^*_{\ell_{\exp}}(\overline{\mathcal{H}}, x)$$
$$\geq \mathcal{C}_{\ell_{\exp}}(\overline{h}, x) - \inf_{\mu \in [-\mathcal{S}(x, y_{\max}), \mathcal{S}(x, \overline{h}(x))]} \mathcal{C}_{\ell_{\exp}}(\overline{h}_\mu, x)$$
$$= \sup_{\mu \in [-\mathcal{S}(x, y_{\max}), \mathcal{S}(x, \overline{h}(x))]} \left\{ p(x, y_{\max})\left[\frac{1}{\mathcal{S}(x, y_{\max})} - \frac{1}{\mathcal{S}(x, \overline{h}(x)) - \mu}\right] \right.$$
$$\left. + p(x, \overline{h}(x))\left[\frac{1}{\mathcal{S}(x, \overline{h}(x))} - \frac{1}{\mathcal{S}(x, y_{\max}) + \mu}\right]\right\}.$$

Differentiating with respect to $\mu$ yields the optimal value

$$\mu^* = \frac{\sqrt{p(x, \overline{h}(x))}\mathcal{S}(x, \overline{h}(x)) - \sqrt{p(x, y_{\max})}\mathcal{S}(x, y_{\max})}{\sqrt{p(x, y_{\max})} + \sqrt{p(x, \overline{h}(x))}}.$$

Plugging that value in the inequality gives:

$$\Delta\mathcal{C}_{\ell_{\exp}, \overline{\mathcal{H}}}(\overline{h}, x) \geq \frac{p(x, y_{\max})}{\mathcal{S}(x, y_{\max})} + \frac{p(x, \overline{h}(x))}{\mathcal{S}(x, \overline{h}(x))} - \frac{\left(\sqrt{p(x, y_{\max})} + \sqrt{p(x, \overline{h}(x))}\right)^2}{\mathcal{S}(x, y_{\max}) + \mathcal{S}(x, \overline{h}(x))}.$$

Differentiating with respect to $\mathcal{S}$ to show that the minimum is attained for $\mathcal{S}(x,\overline{h}(x)) = \mathcal{S}(x,y_{\max}) = \frac{1}{2}$, which implies

$$\Delta\mathcal{C}_{\ell_{\exp},\overline{\mathcal{H}}}(\overline{h},x) \geq \left(\sqrt{p(x,y_{\max})} - \sqrt{p(x,\overline{h}(x))}\right)^2$$

$$= \frac{\left(p(x,\overline{h}(x)) - p(x,y_{\max})\right)^2}{\left(\sqrt{p(x,\overline{h}(x))} + \sqrt{p(x,y_{\max})}\right)^2}.$$

By the concavity of the square-root function, for all $a,b \in [0,1]$, we have $\frac{1}{2}\left(\sqrt{a} + \sqrt{b}\right) \leq \sqrt{\frac{1}{2}(a+b)}$, thus we can write

$$\Delta\mathcal{C}_{\ell_{\exp},\overline{\mathcal{H}}}(\overline{h},x) \geq \frac{\left(p(x,\overline{h}(x)) - p(x,y_{\max})\right)^2}{2\left(p(x,\overline{h}(x)) + p(x,y_{\max})\right)}$$

$$\geq \frac{\left(p(x,\overline{h}(x)) - p(x,y_{\max})\right)^2}{2} \qquad (p(x,\overline{h}(x)) + p(x,y_{\max}) \leq 1)$$

$$= \frac{1}{2}\left(\Delta\mathcal{C}_{\ell_{0-1},\overline{\mathcal{H}}}(\overline{h},x)\right)^2. \qquad \text{(by Lemma 8 and } \overline{H}(x) = \mathcal{Y}\cup\{0\})$$

Since the function $\frac{t^2}{2}$ is convex, by Jensen's inequality, we obtain for any hypothesis $\overline{h} \in \overline{\mathcal{H}}$ and any distribution,

$$\frac{\left(\mathbb{E}_X\left[\Delta\mathcal{C}_{\ell_{0-1},\overline{\mathcal{H}}}(\overline{h},x)\right]\right)^2}{2} \leq \mathbb{E}_X\left[\frac{\Delta\mathcal{C}_{\ell_{0-1},\overline{\mathcal{H}}}(\overline{h},x)^2}{2}\right] \leq \mathbb{E}_X\left[\Delta\mathcal{C}_{\ell_{\exp},\overline{\mathcal{H}}}(\overline{h},x)\right],$$

which leads to

$$\mathcal{E}_{\ell_{0-1}}(\overline{h}) - \mathcal{E}^*_{\ell_{0-1}}(\overline{\mathcal{H}}) \leq \sqrt{2}\left(\mathcal{E}_{\ell_{\exp}}(\overline{h}) - \mathcal{E}^*_{\ell_{\exp}}(\overline{\mathcal{H}}) + \mathcal{M}_{\ell_{\exp}}(\overline{\mathcal{H}})\right)^{\frac{1}{2}} - \mathcal{M}_{\ell_{0-1}}(\overline{\mathcal{H}}).$$

$\square$

### E.3 Generalized cross-entropy loss

**Theorem 11** ($\overline{\mathcal{H}}$-**consistency bound for generalized cross-entropy loss**). *Assume that $\mathcal{H}$ is symmetric and complete. Then, for any $\lambda \in \mathbb{R}$, hypothesis $\overline{h} \in \overline{\mathcal{H}}$ and any distribution,*

$$\mathcal{E}_{\ell_{0-1}}(\overline{h}) - \mathcal{E}^*_{\ell_{0-1}}(\overline{\mathcal{H}}) \leq \sqrt{2(n+1)^\alpha}\left(\mathcal{E}_{\ell_{\gce}}(\overline{h}) - \mathcal{E}^*_{\ell_{\gce}}(\overline{\mathcal{H}}) + \mathcal{M}_{\ell_{\gce}}(\overline{\mathcal{H}})\right)^{\frac{1}{2}} - \mathcal{M}_{\ell_{0-1}}(\overline{\mathcal{H}}).$$

*Proof.* For the generalized cross-entropy loss $\ell_{\gce}$, the conditional $\ell_{\gce}$-risk can be expressed as follows:

$$\mathcal{C}_{\ell_{\gce}}(\overline{h},x)) = \sum_{y\in\mathcal{Y}\cup\{0\}} p(x,y)\frac{1}{\alpha}\left[1 - \left[\frac{e^{\overline{h}(x,y)}}{\sum_{y'\in\mathcal{Y}\cup 0} e^{\overline{h}(x,y')}}\right]^\alpha\right] = \frac{1}{\alpha}\sum_{y\in\mathcal{Y}\cup\{0\}} p(x,y)(1 - \mathcal{S}(x,y)^\alpha)$$

where we let $\mathcal{S}(x,y) = \frac{e^{\overline{h}(x,y)}}{\sum_{y'\in\mathcal{Y}\cup\{0\}} e^{\overline{h}(x,y')}} \in [0,1]$ for any $y \in \mathcal{Y} \cup \{0\}$ with the constraint that $\sum_{y\in\mathcal{Y}\cup\{0\}} \mathcal{S}(x,y) = 1$. Let $y_{\max} = \mathrm{argmax}_{y\in\mathcal{Y}\cup\{0\}} p(x,y)$, where we choose the label with the same deterministic strategy for breaking ties as that of $\overline{h}(x)$. For any $\overline{h} \in \mathcal{H}$ such that $\overline{h}(x) \neq y_{\max}$ and $x \in \mathcal{X}$, by the symmetry and completeness of $\mathcal{H}$, we can always find a family of hypotheses $\left\{\overline{h}_\mu : \mu \in \left[-\mathcal{S}(x,y_{\max}), \mathcal{S}(x,\overline{h}(x))\right]\right\} \subset \overline{\mathcal{H}}$ such that $\mathcal{S}_\mu(x,\cdot) = \frac{e^{\overline{h}_\mu(x,\cdot)}}{\sum_{y'\in\mathcal{Y}\cup\{0\}} e^{\overline{h}_\mu(x,y')}}$ take the following values:

$$\mathcal{S}_\mu(x,y) = \begin{cases} \mathcal{S}(x,y) & \text{if } y \notin \{y_{\max}, \overline{h}(x)\} \\ \mathcal{S}(x,y_{\max}) + \mu & \text{if } y = \overline{h}(x) \\ \mathcal{S}(x,\overline{h}(x)) - \mu & \text{if } y = y_{\max}. \end{cases}$$

Note that $\mathcal{S}_\mu$ satisfies the constraint:

$$\sum_{y \in \mathcal{Y} \cup \{0\}} \mathcal{S}_\mu(x,y) = \sum_{y \in \mathcal{Y} \cup \{0\}} \mathcal{S}(x,y) = 1, \ \forall \mu \in \left[-\mathcal{S}(x, y_{\max}), \mathcal{S}(x, \overline{h}(x))\right].$$

Let $\overline{h} \in \overline{\mathcal{H}}$ be a hypothesis such that $\overline{h}(x) \neq y_{\max}$. By the definition and using the fact that $\overline{H}(x) = \mathcal{Y} \cup \{0\}$ when $\mathcal{H}$ is symmetric, we obtain

$$\Delta\mathcal{C}_{\ell_{\mathrm{gce}},\overline{\mathcal{H}}}(\overline{h}, x)$$
$$= \mathcal{C}_{\ell_{\mathrm{gce}}}(\overline{h}, x) - \mathcal{C}^*_{\ell_{\mathrm{gce}}}(\overline{\mathcal{H}}, x)$$
$$\geq \mathcal{C}_{\ell_{\mathrm{gce}}}(\overline{h}, x) - \inf_{\mu \in [-\mathcal{S}(x, y_{\max}), \mathcal{S}(x, \overline{h}(x))]} \mathcal{C}_{\ell_{\mathrm{gce}}}(\overline{h}_\mu, x)$$
$$= \frac{1}{\alpha} \sup_{\mu \in [-\mathcal{S}(x, y_{\max}), \mathcal{S}(x, \overline{h}(x))]} \left\{ p(x, y_{\max})\left[-\mathcal{S}(x, y_{\max})^\alpha + \left(\mathcal{S}(x, \overline{h}(x)) - \mu\right)^\alpha\right] \right.$$
$$\left. + p(x, \overline{h}(x))\left[-\mathcal{S}(x, \overline{h}(x))^\alpha + \left(\mathcal{S}(x, y_{\max}) + \mu\right)^\alpha\right] \right\}.$$

Differentiating with respect to $\mu$ yields the optimal value

$$\mu^* = \frac{p(x, \overline{h}(x))^{\frac{1}{1-\alpha}} \mathcal{S}(x, \overline{h}(x)) - p(x, y_{\max})^{\frac{1}{1-\alpha}} \mathcal{S}(x, y_{\max})}{p(x, y_{\max})^{\frac{1}{1-\alpha}} + p(x, \overline{h}(x))^{\frac{1}{1-\alpha}}}.$$

Plugging that value in the inequality gives:

$$\Delta\mathcal{C}_{\ell_{\mathrm{gce}},\overline{\mathcal{H}}}(\overline{h}, x) \geq \frac{1}{\alpha}\left(\mathcal{S}(x, \overline{h}(x)) + \mathcal{S}(x, y_{\max})\right)^\alpha \left(p(x, y_{\max})^{\frac{1}{1-\alpha}} + p(x, \overline{h}(x))^{\frac{1}{1-\alpha}}\right)^{1-\alpha}$$
$$- \frac{1}{\alpha}p(x, y_{\max})\mathcal{S}(x, y_{\max})^\alpha - \frac{1}{\alpha}p(x, \overline{h}(x))\mathcal{S}(x, \overline{h}(x))^\alpha.$$

Differentiating with respect to $\mathcal{S}$ to show that the minimum is attained for $\mathcal{S}(x, \overline{h}(x)) = \mathcal{S}(x, y_{\max}) = \frac{1}{n+1}$, which implies

$$\Delta\mathcal{C}_{\ell_{\mathrm{gce}},\overline{\mathcal{H}}}(\overline{h}, x) \geq \frac{1}{\alpha(n+1)^\alpha}\left[2^\alpha\left(p(x, y_{\max})^{\frac{1}{1-\alpha}} + p(x, \overline{h}(x))^{\frac{1}{1-\alpha}}\right)^{1-\alpha} - p(x, y_{\max}) - p(x, \overline{h}(x))\right].$$

By using the fact that for all $a, b \in [0,1]$, $0 \leq a + b \leq 1$, we have $\left(\frac{a^{\frac{1}{1-\alpha}} + b^{\frac{1}{1-\alpha}}}{2}\right)^{1-\alpha} - \frac{a+b}{2} \geq \frac{\alpha}{4}(a-b)^2$, thus we can write

$$\Delta\mathcal{C}_{\ell_{\mathrm{gce}},\overline{\mathcal{H}}}(\overline{h}, x) \geq \frac{\left(p(x, \overline{h}(x)) - p(x, y_{\max})\right)^2}{2(n+1)^\alpha}$$
$$= \frac{1}{2(n+1)^\alpha}\left(\Delta\mathcal{C}_{\ell_{0-1},\overline{\mathcal{H}}}(\overline{h}, x)\right)^2. \qquad \text{(by Lemma 8 and } \overline{H}(x) = \mathcal{Y} \cup \{0\}\text{)}$$

Since the function $\frac{t^2}{2(n+1)^\alpha}$ is convex, by Jensen's inequality, we obtain for any hypothesis $\overline{h} \in \overline{\mathcal{H}}$ and any distribution,

$$\frac{\left(\mathbb{E}_X\left[\Delta\mathcal{C}_{\ell_{0-1},\overline{\mathcal{H}}}(\overline{h}, x)\right]\right)^2}{2(n+1)^\alpha} \leq \mathbb{E}_X\left[\frac{\Delta\mathcal{C}_{\ell_{0-1},\overline{\mathcal{H}}}(\overline{h}, x)^2}{2(n+1)^\alpha}\right] \leq \mathbb{E}_X\left[\Delta\mathcal{C}_{\ell_{\mathrm{gce}},\overline{\mathcal{H}}}(\overline{h}, x)\right]$$

which leads to

$$\mathcal{E}_{\ell_{0-1}}(\overline{h}) - \mathcal{E}^*_{\ell_{0-1}}(\overline{\mathcal{H}}) \leq \sqrt{2(n+1)^\alpha}\left(\mathcal{E}_{\ell_{\mathrm{gce}}}(\overline{h}) - \mathcal{E}^*_{\ell_{\mathrm{gce}}}(\overline{\mathcal{H}}) + \mathcal{M}_{\ell_{\mathrm{gce}}}(\overline{\mathcal{H}})\right)^{\frac{1}{2}} - \mathcal{M}_{\ell_{0-1}}(\overline{\mathcal{H}}).$$

$\square$

### E.4 Mean absolute error loss

**Theorem 12** ($\overline{\mathcal{H}}$-**consistency bound for mean absolute error loss**). *Assume that $\mathcal{H}$ is symmetric and complete. Then, for any $\lambda \in \mathbb{R}$, hypothesis $\overline{h} \in \overline{\mathcal{H}}$ and any distribution,*

$$\mathcal{E}_{\ell_{0-1}}(\overline{h}) - \mathcal{E}^*_{\ell_{0-1}}(\overline{\mathcal{H}}) \leq (n+1)\big(\mathcal{E}_{\ell_{\text{mae}}}(\overline{h}) - \mathcal{E}^*_{\ell_{\text{mae}}}(\overline{\mathcal{H}}) + \mathcal{M}_{\ell_{\text{mae}}}(\overline{\mathcal{H}})\big) - \mathcal{M}_{\ell_{0-1}}(\overline{\mathcal{H}}).$$

*Proof.* For the mean absolute error loss $\ell_{\text{mae}}$, the conditional $\ell_{\text{mae}}$-risk can be expressed as follows:

$$\mathcal{C}_{\ell_{\text{mae}}}(\overline{h}, x)) = \sum_{y \in \mathcal{Y} \cup \{0\}} p(x, y)\left(1 - \frac{e^{\overline{h}(x,y)}}{\sum_{y' \in \mathcal{Y} \cup 0} e^{\overline{h}(x,y')}}\right) = \sum_{y \in \mathcal{Y} \cup \{0\}} p(x, y)(1 - \mathcal{S}(x, y))$$

where we let $\mathcal{S}(x, y) = \frac{e^{\overline{h}(x,y)}}{\sum_{y' \in \mathcal{Y} \cup \{0\}} e^{\overline{h}(x,y')}} \in [0, 1]$ for any $y \in \mathcal{Y} \cup \{0\}$ with the constraint that $\sum_{y \in \mathcal{Y} \cup \{0\}} \mathcal{S}(x, y) = 1$. Let $y_{\max} = \text{argmax}_{y \in \mathcal{Y} \cup \{0\}} p(x, y)$, where we choose the label with the same deterministic strategy for breaking ties as that of $\overline{h}(x)$. For any $\overline{h} \in \mathcal{H}$ such that $\overline{h}(x) \neq y_{\max}$ and $x \in \mathcal{X}$, by the symmetry and completeness of $\mathcal{H}$, we can always find a family of hypotheses $\{\overline{h}_\mu : \mu \in [-\mathcal{S}(x, y_{\max}), \mathcal{S}(x, \overline{h}(x))]\} \subset \overline{\mathcal{H}}$ such that $\mathcal{S}_\mu(x, \cdot) = \frac{e^{\overline{h}_\mu(x,\cdot)}}{\sum_{y' \in \mathcal{Y} \cup \{0\}} e^{\overline{h}_\mu(x,y')}}$ take the following values:

$$\mathcal{S}_\mu(x, y) = \begin{cases} \mathcal{S}(x, y) & \text{if } y \notin \{y_{\max}, \overline{h}(x)\} \\ \mathcal{S}(x, y_{\max}) + \mu & \text{if } y = \overline{h}(x) \\ \mathcal{S}(x, \overline{h}(x)) - \mu & \text{if } y = y_{\max}. \end{cases}$$

Note that $\mathcal{S}_\mu$ satisfies the constraint:

$$\sum_{y \in \mathcal{Y} \cup \{0\}} \mathcal{S}_\mu(x, y) = \sum_{y \in \mathcal{Y} \cup \{0\}} \mathcal{S}(x, y) = 1, \ \forall \mu \in [-\mathcal{S}(x, y_{\max}), \mathcal{S}(x, \overline{h}(x))].$$

Let $\overline{h} \in \overline{\mathcal{H}}$ be a hypothesis such that $\overline{h}(x) \neq y_{\max}$. By the definition and using the fact that $\overline{H}(x) = \mathcal{Y} \cup \{0\}$ when $\mathcal{H}$ is symmetric, we obtain

$$\Delta\mathcal{C}_{\ell_{\text{mae}}, \overline{\mathcal{H}}}(\overline{h}, x)$$
$$= \mathcal{C}_{\ell_{\text{mae}}}(\overline{h}, x) - \mathcal{C}^*_{\ell_{\text{mae}}}(\overline{\mathcal{H}}, x)$$
$$\geq \mathcal{C}_{\ell_{\text{mae}}}(\overline{h}, x) - \inf_{\mu \in [-\mathcal{S}(x, y_{\max}), \mathcal{S}(x, \overline{h}(x))]} \mathcal{C}_{\ell_{\text{mae}}}(\overline{h}_\mu, x)$$
$$= \sup_{\mu \in [-\mathcal{S}(x, y_{\max}), \mathcal{S}(x, \overline{h}(x))]} \left\{ p(x, y_{\max})\big[-\mathcal{S}(x, y_{\max}) + \mathcal{S}(x, \overline{h}(x)) - \mu\big] \right.$$
$$\left. + p(x, \overline{h}(x))\big[-\mathcal{S}(x, \overline{h}(x)) + \mathcal{S}(x, y_{\max}) + \mu\big] \right\}.$$

Differentiating with respect to $\mu$ yields the optimum value $\mu^* = -\mathcal{S}(x, y_{\max})$. Plugging that value in the inequality gives:

$$\Delta\mathcal{C}_{\ell_{\text{mae}}, \overline{\mathcal{H}}}(\overline{h}, x) \geq p(x, y_{\max})\mathcal{S}(x, \overline{h}(x)) - p(x, \overline{h}(x))\mathcal{S}(x, \overline{h}(x)).$$

Differentiating with respect to $\mathcal{S}$ to show that the minimum is attained for $\mathcal{S}(x, \overline{h}(x)) = \frac{1}{n+1}$, which implies

$$\Delta\mathcal{C}_{\ell_{\text{mae}}, \overline{\mathcal{H}}}(\overline{h}, x) \geq \frac{1}{n+1}\big(p(x, y_{\max}) - p(x, \overline{h}(x))\big)$$
$$= \frac{1}{n+1}\big(\Delta\mathcal{C}_{\ell_{0-1}, \overline{\mathcal{H}}}(\overline{h}, x)\big). \qquad \text{(by Lemma 8 and } \overline{H}(x) = \mathcal{Y} \cup \{0\})$$

Therefore, we obtain for any hypothesis $\overline{h} \in \overline{\mathcal{H}}$ and any distribution,

$$\frac{\mathbb{E}_X\big[\Delta\mathcal{C}_{\ell_{0-1}, \overline{\mathcal{H}}}(\overline{h}, x)\big]}{n+1} \leq \mathbb{E}_X\big[\Delta\mathcal{C}_{\ell_{\text{mae}}, \overline{\mathcal{H}}}(\overline{h}, x)\big],$$

which leads to

$$\mathcal{E}_{\ell_{0-1}}(\overline{h}) - \mathcal{E}^*_{\ell_{0-1}}(\overline{\mathcal{H}}) \leq (n+1)\big(\mathcal{E}_{\ell_{\text{mae}}}(\overline{h}) - \mathcal{E}^*_{\ell_{\text{mae}}}(\overline{\mathcal{H}}) + \mathcal{M}_{\ell_{\text{mae}}}(\overline{\mathcal{H}})\big) - \mathcal{M}_{\ell_{0-1}}(\overline{\mathcal{H}}).$$

$\square$

## F  Proof of realizable consistency for score-based two-stage surrogate losses (Theorem 5)

**Theorem 5** (**Realizable $\mathcal{H}$-consistency for score-based two-stage surrogates**). *Assume that $\mathcal{H}$ is closed under scaling and $c_j(x, y) = \beta_j, \forall (x, y) \in \mathcal{X} \times \mathcal{Y}$. Let $\ell_1$ and $\ell_2$ be the logistic loss. Let $\hat{h}_p$ be the minimizer of $\mathcal{E}_{\ell_1}$ and $\hat{h}_d$ be the minimizer of $\mathcal{E}_{\mathsf{L}_{\hat{h}_p}}$ such that $\mathcal{E}_{\mathsf{L}_{\hat{h}_p}}(\hat{h}_d) = \min_h \mathcal{E}_{\mathsf{L}_{h_p}}(h_d)$. Then, the following equality holds for any $(\mathcal{H}, \mathcal{R})$-realizable distribution,*

$$\mathcal{E}_{\mathsf{L}_{\mathrm{def}}}(\hat{h}) = 0, \ where \ \hat{h} = (\hat{h}_p, \hat{h}_d).$$

*Proof.* First, by definition, it is straightforward to see that for any $h, x, y$, $\mathsf{L}_{h_p}(h_d, x, y)$ upper bounds the deferral loss $\mathsf{L}_{\mathrm{def}}$. Consider a data distribution and costs under which there exists $h^* \in \mathcal{H}$ such that $\mathcal{E}_{\mathsf{L}_{\mathrm{def}}}(h^*) = 0$.

Let $\hat{h}_p$ be the minimizer of $\mathcal{E}_{\ell_1}$ and $\hat{h}_d$ the minimizer of $\mathcal{E}_{\mathsf{L}_{\hat{h}_p}}$ Then, using the fact that $\mathsf{L}_h$ upper bounds the deferral loss $\mathsf{L}_{\mathrm{def}}$, we have $\mathcal{E}_{\mathsf{L}_{\mathrm{def}}}(\hat{h}) \le \mathcal{E}_{\mathsf{L}_{\hat{h}_p}}(\hat{h}_d)$.

Next we analyze two cases. If for a point $x$, deferral occurs, that is there exists $j^* \in [n_e]$, such that $\mathsf{h}^*(x) = n + j^*$, then we must have $c_{j^*} = 0$ for all $x$ since the data is realizable and $c_{j^*}$ is constant. Therefore, there exists an optimal $h^{**}$ deferring all the points to the $j^*$th expert. Then, by the assumption that $\mathcal{H}$ is closed under scaling and the Lebesgue dominated convergence theorem, for $\ell_2$ being the logistic loss, $\mathcal{E}_{\mathsf{L}_{\mathrm{def}}}(\hat{h}) \le \mathcal{E}_{\mathsf{L}_{\hat{h}_p}}(\hat{h}_d) \le \lim_{\tau \to +\infty} \mathcal{E}_{\mathsf{L}_{h_p^{**}}}(\tau h_d^{**}) = 0$, where we used the fact that in the limit of $\tau \to +\infty$ the logistic loss term $\ell_2(\overline{h}_d^{**}, x, j)$ corresponding to $j \ne j^*$ is zero.

On the other hand, if no deferral occurs for any point, that is $\mathsf{h}^*(x) \in [n]$ for any $x$, then we must have $\mathbb{1}_{\mathsf{h}_p^*(x) \ne y} = 0$ for all $(x, y)$ since the data is realizable. Using the fact that $\mathcal{H}$ is closed under scaling and that the logistic loss is realizable $\mathcal{H}$-consistent in the standard classification, we obtain $\mathbb{1}_{\hat{h}_p(x) \ne y} = 0$ for all $(x, y)$. Then, by the assumption that $\mathcal{H}$ is closed under scaling and the Lebesgue dominated convergence theorem, for $\ell_2$ being the logistic loss, $\mathcal{E}_{\mathsf{L}_{\mathrm{def}}}(\hat{h}) \le \mathcal{E}_{\mathsf{L}_{\hat{h}_p}}(\hat{h}_d) \le \lim_{\tau \to +\infty} \mathcal{E}_{\mathsf{L}_{h_p^*}}(\tau h_d^*) = 0$, where we used the fact that in the limit of $\tau \to +\infty$ the logistic loss term $\ell_2(\overline{h}_d^*, x, j)$ corresponding to $j \ne 0$ is zero.

Therefore, the optimal solution from minimizing score-based two-stage surrogates leads to a zero error solution of the deferral loss, which proves that the score-based two-stage surrogate loss is realizable consistent. □

## G  Proof of $(\mathcal{H}, \mathcal{R})$-consistency bounds for predictor-rejector two-stage surrogate losses (Theorem 6)

**Theorem 6** ($(\mathcal{H}, \mathcal{R})$-**consistency bounds for predictor-rejector two-stage surrogates**). *Assume that $\ell_1$ admits an $\mathcal{H}$-consistency bound and $\ell_2$ admits an $\overline{\mathcal{R}}$-consistency bound with respect to the multi-class zero-one classification loss $\ell_{0-1}$ respectively. Thus, there are non-decreasing concave functions $\Gamma_1$ and $\Gamma_2$ such that, for all $h \in \mathcal{H}$ and $\overline{r} \in \overline{\mathcal{R}}$, we have*

$$\mathcal{E}_{\ell_{0-1}}(h) - \mathcal{E}_{\ell_{0-1}}^*(\mathcal{H}) + \mathcal{M}_{\ell_{0-1}}(\mathcal{H}) \le \Gamma_1\big(\mathcal{E}_{\ell_1}(h) - \mathcal{E}_{\ell_1}^*(\mathcal{H}) + \mathcal{M}_{\ell_1}(\mathcal{H})\big)$$

$$\mathcal{E}_{\ell_{0-1}}(\overline{r}) - \mathcal{E}_{\ell_{0-1}}^*(\overline{\mathcal{R}}) + \mathcal{M}_{\ell_{0-1}}(\overline{\mathcal{R}}) \le \Gamma_2\big(\mathcal{E}_{\ell_2}(\overline{r}) - \mathcal{E}_{\ell_2}^*(\overline{\mathcal{R}}) + \mathcal{M}_{\ell_2}(\overline{\mathcal{R}})\big).$$

*Then, the following holds for all $h \in \mathcal{H}$ and $r \in \mathcal{R}$:*

$$\mathcal{E}_{\mathsf{L}_{\mathrm{def}}}(h, r) - \mathcal{E}_{\mathsf{L}_{\mathrm{def}}}^*(\mathcal{H}, \mathcal{R}) + \mathcal{M}_{\mathsf{L}_{\mathrm{def}}}(\mathcal{H}, \mathcal{R})$$

$$\le \Gamma_1\big(\mathcal{E}_{\ell_1}(h) - \mathcal{E}_{\ell_1}^*(\mathcal{H}) + \mathcal{M}_{\ell_1}(\mathcal{H})\big) + \left(1 + \sum_{j=1}^{n_e} \overline{c}_j\right)\Gamma_2\left(\frac{\mathcal{E}_{\mathsf{L}_h}(r) - \mathcal{E}_{\mathsf{L}_h}^*(\mathcal{R}) + \mathcal{M}_{\mathsf{L}_h}(\mathcal{R})}{\sum_{j=1}^{n_e} \underline{c}_j}\right),$$

*where the constant factors $\big(1 + \sum_{j=1}^{n_e} \overline{c}_j\big)$ and $\frac{1}{\sum_{j=1}^{n_e} \underline{c}_j}$ can be removed when $\Gamma_2$ is linear.*

*Proof.* By definition,

$$\mathsf{L}_{\mathrm{def}}(h, r, x, y) = \mathbb{1}_{\mathsf{h}(x)\neq y}\mathbb{1}_{\mathsf{r}(x)=0} + \sum_{j=1}^{n_e} c_j(x,y)\mathbb{1}_{\mathsf{r}(x)=j}.$$

Let $\bar{c}_0(x,y) = \mathbb{1}_{\mathsf{h}(x)=y}$. We can rewrite $\mathcal{E}_{\mathsf{L}_{\mathrm{def}}}(h,r) - \mathcal{E}^*_{\mathsf{L}_{\mathrm{def}}}(\mathcal{H},\mathcal{R}) + \mathcal{M}_{\mathsf{L}_{\mathrm{def}}}(\mathcal{H},\mathcal{R})$ as

$$
\begin{aligned}
&\mathcal{E}_{\mathsf{L}_{\mathrm{def}}}(h,r) - \mathcal{E}^*_{\mathsf{L}_{\mathrm{def}}}(\mathcal{H},\mathcal{R}) + \mathcal{M}_{\mathsf{L}_{\mathrm{def}}}(\mathcal{H},\mathcal{R})\\
&= \mathbb{E}_X\big[\mathcal{C}_{\mathsf{L}_{\mathrm{def}}}(h,r,x) - \mathcal{C}^*_{\mathsf{L}_{\mathrm{def}}}(\mathcal{H},\mathcal{R},x)\big]\\
&= \mathbb{E}_X\Big[\mathcal{C}_{\mathsf{L}_{\mathrm{def}}}(h,r,x) - \inf_{r\in\mathcal{R}}\mathcal{C}_{\mathsf{L}_{\mathrm{def}}}(h,r,x) + \inf_{r\in\mathcal{R}}\mathcal{C}_{\mathsf{L}_{\mathrm{def}}}(h,r,x) - \mathcal{C}^*_{\mathsf{L}_{\mathrm{def}}}(\mathcal{H},\mathcal{R},x)\Big]\\
&= \mathbb{E}_X\Big[\mathcal{C}_{\mathsf{L}_{\mathrm{def}}}(h,r,x) - \inf_{r\in\mathcal{R}}\mathcal{C}_{\mathsf{L}_{\mathrm{def}}}(h,r,x)\Big] + \mathbb{E}_X\Big[\inf_{r\in\mathcal{R}}\mathcal{C}_{\mathsf{L}_{\mathrm{def}}}(h,r,x) - \mathcal{C}^*_{\mathsf{L}_{\mathrm{def}}}(\mathcal{H},\mathcal{R},x)\Big]
\end{aligned}
\tag{7}
$$

Let $\bar{p}(x,j) = \frac{\mathbb{E}_y[\bar{c}_j(x,y)]}{\sum_{j=0}^{n_e}\mathbb{E}_y[\bar{c}_j(x,y)]}$ for any $j \in \{0,\dots,n_e\}$. Note that $\bar{p}(x,\cdot)$ is the probability vector on the label space $\{0,\dots,n_e\}$. For any $r \in \mathcal{R}$, we define $\bar{r}$ as its augmented hypothesis: $\bar{r}(x,0) = 0, \bar{r}(x,1) = -r_1(x),\dots,\bar{r}(x,n_e) = -r_{n_e}(x)$. By the assumptions, we have

$$
\begin{aligned}
&\mathcal{C}_{\mathsf{L}_{\mathrm{def}}}(h,r,x) - \inf_{r\in\mathcal{R}}\mathcal{C}_{\mathsf{L}_{\mathrm{def}}}(h,r,x)\\
&= \mathbb{E}_y\Big[\mathbb{1}_{\mathsf{h}(x)\neq y}\mathbb{1}_{\mathsf{r}(x)=0} + \sum_{j=1}^{n_e}c_j(x,y)\mathbb{1}_{\mathsf{r}(x)=j}\Big] - \inf_{r\in\mathcal{R}}\mathbb{E}_y\Big[\mathbb{1}_{\mathsf{h}(x)\neq y}\mathbb{1}_{\mathsf{r}(x)=0} + \sum_{j=1}^{n_e}c_j(x,y)\mathbb{1}_{\mathsf{r}(x)=j}\Big]\\
&= \mathbb{E}_y\Big[\sum_{j=0}^{n_e}\bar{c}_j(x,y)\Big]\times\Big[\sum_{j=0}^{n_e}\bar{p}(x,j)\ell_{0-1}(\bar{r},x,j) - \inf_{\bar{r}\in\mathcal{R}}\sum_{j=0}^{n_e}\bar{p}(x,j)\ell_{0-1}(\bar{r},x,j)\Big]\\
&\leq \mathbb{E}_y\Big[\sum_{j=0}^{n_e}\bar{c}_j(x,y)\Big]\times\Gamma_2\Big[\sum_{j=0}^{n_e}\bar{p}(x,j)\ell_2(\bar{r},x,j) - \inf_{\bar{r}\in\mathcal{R}}\sum_{j=0}^{n_e}\bar{p}(x,j)\ell_2(\bar{r},x,j)\Big]
\end{aligned}
$$

$$\text{(By } \overline{\mathcal{R}}\text{-consistency bounds of } \ell_2 \text{ under assumption)}$$

$$
= \mathbb{E}_y\Big[\sum_{j=0}^{n_e}\bar{c}_j(x,y)\Big]\Gamma_2\left(\frac{\mathbb{E}_y[\mathsf{L}_h(r,x,y)] - \inf_{r\in\mathcal{R}}\mathbb{E}_y[\mathsf{L}_h(r,x,y)]}{\mathbb{E}_y\big[\sum_{j=0}^{n_e}\bar{c}_j(x,y)\big]}\right)
$$

$$\left(\bar{p}(x,j) = \frac{\mathbb{E}_y[\bar{c}_j(x,y)]}{\sum_{j=0}^{n_e}\mathbb{E}_y[\bar{c}_j(x,y)]} \text{ and formulation (5)}\right)$$

$$
\leq \begin{cases}
\Gamma_2\big(\mathcal{C}_{\mathsf{L}_h}(r,x) - \mathcal{C}^*_{\mathsf{L}_h}(\mathcal{R},x)\big) & \text{when } \Gamma_2 \text{ is linear}\\
\big(1 + \sum_{j=1}^{n_e}\bar{c}_j\big)\Gamma_2\Big(\frac{\mathcal{C}_{\mathsf{L}_h}(r,x)-\mathcal{C}^*_{\mathsf{L}_h}(\mathcal{R},x)}{\sum_{j=1}^{n_e}\underline{c}_j}\Big) & \text{otherwise}
\end{cases}
$$

$$\Big(\textstyle\sum_{j=1}^{n_e}\underline{c}_j \leq \mathbb{E}_y\big[\sum_{j=0}^{n_e}\bar{c}_j(x,y)\big] \leq 1 + \sum_{j=1}^{n_e}\bar{c}_j \text{ and } \Gamma_2 \text{ is non-decreasing}\Big)$$

$$
= \begin{cases}
\Gamma_2(\Delta\mathcal{C}_{\mathsf{L}_h,\mathcal{R}}(r,x)) & \text{when } \Gamma_2 \text{ is linear}\\
\big(1 + \sum_{j=1}^{n_e}\bar{c}_j\big)\Gamma_2\Big(\frac{\Delta\mathcal{C}_{\mathsf{L}_h,\mathcal{R}}(r,x)}{\sum_{j=1}^{n_e}\underline{c}_j}\Big) & \text{otherwise}
\end{cases}
$$

and

$$
\begin{aligned}
&\inf_{r\in\mathcal{R}}\mathcal{C}_{\mathsf{L}_{\mathrm{def}}}(h,r,x) - \mathcal{C}^*_{\mathsf{L}_{\mathrm{def}}}(\mathcal{H},\mathcal{R},x)\\
&= \inf_{r\in\mathcal{R}}\mathcal{C}_{\mathsf{L}_{\mathrm{def}}}(h,r,x) - \inf_{h\in\mathcal{H},r\in\mathcal{R}}\mathcal{C}_{\mathsf{L}_{\mathrm{def}}}(h,r,x)\\
&= \inf_{r\in\mathcal{R}}\mathbb{E}_y\Big[\mathbb{1}_{\mathsf{h}(x)\neq y}\mathbb{1}_{\mathsf{r}(x)=0} + \sum_{j=1}^{n_e}c_j(x,y)\mathbb{1}_{\mathsf{r}(x)=j}\Big] - \inf_{h\in\mathcal{H},r\in\mathcal{R}}\mathbb{E}_y\Big[\mathbb{1}_{\mathsf{h}(x)\neq y}\mathbb{1}_{\mathsf{r}(x)=0} + \sum_{j=1}^{n_e}c_j(x,y)\mathbb{1}_{\mathsf{r}(x)=j}\Big]\\
&= \min\Big\{\mathbb{E}_y\big[\mathbb{1}_{\mathsf{h}(x)\neq y}\big], \mathbb{E}_y[c_j(x,y)]\Big\} - \min\Big\{\inf_{h\in\mathcal{H}}\mathbb{E}_y\big[\mathbb{1}_{\mathsf{h}(x)\neq y}\big], \mathbb{E}_y[c_j(x,y)]\Big\}\\
&\leq \mathbb{E}_y\big[\mathbb{1}_{\mathsf{h}(x)\neq y}\big] - \inf_{h\in\mathcal{H}}\mathbb{E}_y\big[\mathbb{1}_{\mathsf{h}(x)\neq y}\big]\\
&= \mathcal{C}_{\ell_{0-1}}(h,x) - \mathcal{C}^*_{\ell_{0-1}}(\mathcal{H},x)\\
&= \Delta\mathcal{C}_{\ell_{0-1}}(h,x)\\
&\leq \Gamma_1(\Delta\mathcal{C}_\ell(h,x)). \qquad\qquad\qquad\qquad \text{(By } \mathcal{H}\text{-consistency bounds of } \ell \text{ under assumption)}
\end{aligned}
$$

Therefore, by (7), we obtain

$$
\mathcal{E}_{\mathsf{L}_{\mathrm{def}}}(h, r) - \mathcal{E}^*_{\mathsf{L}_{\mathrm{def}}}(\mathcal{H}, \mathcal{R}) + \mathcal{M}_{\mathsf{L}_{\mathrm{def}}}(\mathcal{H}, \mathcal{R})
$$

$$
\leq \begin{cases}
\mathbb{E}_X\big[\Gamma_2(\Delta\mathcal{C}_{\mathsf{L}_h, \mathcal{R}}(r, x))\big] + \mathbb{E}_X\big[\Gamma_1(\Delta\mathcal{C}_\ell(h, x))\big] & \text{when } \Gamma_2 \text{ is linear} \\[2mm]
\big(1 + \sum_{j=1}^{n_e} \overline{c}_j\big)\mathbb{E}_X\Big[\Gamma_2\Big(\frac{\Delta\mathcal{C}_{\mathsf{L}_h, \mathcal{R}}(r, x)}{\sum_{j=1}^{n_e} \underline{c}_j}\Big)\Big] + \mathbb{E}_X\big[\Gamma_1(\Delta\mathcal{C}_\ell(h, x))\big] & \text{otherwise}
\end{cases}
$$

$$
\leq \begin{cases}
\Gamma_2\big(\mathbb{E}_X\big[\Gamma_2(\Delta\mathcal{C}_{\mathsf{L}_h, \mathcal{R}}(r, x))\big]\big) + \Gamma_1\big(\mathbb{E}_X\big[\Delta\mathcal{C}_\ell(h, x)\big]\big) & \text{when } \Gamma_2 \text{ is linear} \\[2mm]
\big(1 + \sum_{j=1}^{n_e} \overline{c}_j\big)\Gamma_2\Big(\mathbb{E}_X\Big[\frac{\Delta\mathcal{C}_{\mathsf{L}_h, \mathcal{R}}(r, x)}{\sum_{j=1}^{n_e} \underline{c}_j}\Big]\Big) + \Gamma_1\big(\mathbb{E}_X\big[\Delta\mathcal{C}_\ell(h, x)\big]\big) & \text{otherwise}
\end{cases}
$$

$$
(\Gamma_1 \text{ and } \Gamma_2 \text{ are concave})
$$

$$
= \begin{cases}
\Gamma_1\big(\mathcal{E}_\ell(h) - \mathcal{E}^*_\ell(\mathcal{H}) + \mathcal{M}_\ell(\mathcal{H})\big) + \Gamma_2\big(\mathcal{E}_{\mathsf{L}_h}(r) - \mathcal{E}^*_{\mathsf{L}_h}(\mathcal{R}) + \mathcal{M}_{\mathsf{L}_h}(\mathcal{R})\big) & \text{when } \Gamma_2 \text{ is linear} \\[2mm]
\Gamma_1\big(\mathcal{E}_\ell(h) - \mathcal{E}^*_\ell(\mathcal{H}) + \mathcal{M}_\ell(\mathcal{H})\big) + \big(1 + \sum_{j=1}^{n_e} \overline{c}_j\big)\Gamma_2\Big(\frac{\mathcal{E}_{\mathsf{L}_h}(r) - \mathcal{E}^*_{\mathsf{L}_h}(\mathcal{R}) + \mathcal{M}_{\mathsf{L}_h}(\mathcal{R})}{\sum_{j=1}^{n_e} \underline{c}_j}\Big) & \text{otherwise,}
\end{cases}
$$

which completes the proof. $\qquad\square$

## H  Proof of realizable consistency for predictor-rejector two-stage surrogate losses (Theorem 7)

**Theorem 7** (**Realizable $(\mathcal{H}, \mathcal{R})$-consistency for predictor-rejector two-stage surrogates**). *Assume that $\mathcal{H}$ and $\mathcal{R}$ is closed under scaling and $c_j(x, y) = \beta_j, \forall (x, y) \in \mathcal{X} \times \mathcal{Y}$. Let $\ell_1$ and $\ell_2$ be the logistic loss. Let $\hat{h}$ be the minimizer of $\mathcal{E}_{\ell_1}$ and $\hat{r}$ be the minimizer of $\mathcal{E}_{\mathsf{L}_{\hat{h}}}$. Then, the following holds for any $(\mathcal{H}, \mathcal{R})$-realizable distribution,*

$$
\mathcal{E}_{\mathsf{L}_{\mathrm{def}}}(\hat{h}, \hat{r}) = 0.
$$

*Proof.* First, by definition, it is straightforward to see that for any $h, r, x, y$, $\mathsf{L}_h(r, x, y)$ upper bounds the deferral loss $\mathsf{L}_{\mathrm{def}}$. Consider a data distribution and costs under which there exists $h^* \in \mathcal{H}$ and $r^* \in \mathcal{R}$ such that $\mathcal{E}_{\mathsf{L}_{\mathrm{def}}}(h^*, r^*) = 0$.

Let $\hat{h}$ be the minimizer of $\mathcal{E}_{\ell_1}$ and $\hat{r}$ the minimizer of $\mathcal{E}_{\mathsf{L}_{\hat{h}}}$. Then, using the fact that $\mathsf{L}_h$ upper bounds the deferral loss $\mathsf{L}_{\mathrm{def}}$, we have $\mathcal{E}_{\mathsf{L}_{\mathrm{def}}}(\hat{h}, \hat{r}) \leq \mathcal{E}_{\mathsf{L}_{\hat{h}}}(\hat{r})$.

Next we analyze two cases. If for a point $x$, deferral occurs, that is there exists $j^* \in [n_e]$, such that $r^*(x) = j^*$, then we must have $c_{j^*} = 0$ for all $x$ since the data is realizable and $c_{j^*}$ is constant. Therefore, there exists an optimal $r^{**}$ deferring all the points to the $j^*$th expert. Then, by the assumption that $\mathcal{R}$ is closed under scaling and the Lebesgue dominated convergence theorem, for $\ell_2$ being the logistic loss, $\mathcal{E}_{\mathsf{L}_{\mathrm{def}}}(\hat{h}, \hat{r}) \leq \mathcal{E}_{\mathsf{L}_{\hat{h}}}(\hat{r}) \leq \lim_{\tau \to +\infty} \mathcal{E}_{\mathsf{L}_{\hat{h}}}(\tau r^{**}) = 0$, where we used the fact that in the limit of $\tau \to +\infty$ the logistic loss term $\ell_2(\overline{r}^{**}, x, j)$ corresponding to $j \neq j^*$ is zero.

On the other hand, if no deferral occurs for any point, that is $r^*(x) = 0$ for any $x$, then we must have $\mathbb{1}_{h^*(x) \neq y} = 0$ for all $(x, y)$ since the data is realizable. Using the fact that $\mathcal{H}$ is closed under scaling and that the logistic loss is realizable $\mathcal{H}$-consistent in the standard classification, we obtain $\mathbb{1}_{\hat{h}(x) \neq y} = 0$ for all $(x, y)$. Then, by the assumption that $\mathcal{R}$ is closed under scaling and the Lebesgue dominated convergence theorem, for $\ell_2$ being the logistic loss, $\mathcal{E}_{\mathsf{L}_{\mathrm{def}}}(\hat{h}, \hat{r}) \leq \mathcal{E}_{\mathsf{L}_{\hat{h}}}(\hat{r}) \leq \lim_{\tau \to +\infty} \mathcal{E}_{\mathsf{L}_{\hat{h}}}(\tau r^*) = 0$, where we used the fact that in the limit of $\tau \to +\infty$ the logistic loss term $\ell_2(\overline{r}^*, x, j)$ corresponding to $j \neq 0$ is zero.

Therefore, the optimal solution from minimizing predictor-rejector two-stage surrogates leads to a zero error solution of the deferral loss, which proves that the predictor-rejector two-stage surrogate loss is realizable consistent.

$\qquad\square$

