# OpenReview forum: "Two-Stage Learning to Defer with Multiple Experts"
_NeurIPS.cc/2023/Conference — NeurIPS 2023 poster_

### Official Review · Reviewer_Jgi5 · 2023-06-30

**Soundness:** 2 fair
**Presentation:** 3 good
**Contribution:** 3 good
**Rating:** 5
**Confidence:** 4

**Summary:**

This paper studies generalization bounds for learning to defer, with its main focus on "two-stage" algorithms that first fit a classifier using a surrogate classification loss, then separately fit a rejector that decides which expert, if any, to defer the classification to.
By and large, these two-stage algorithms have less theory than _joint_ algorithms that fit the classifier and rejector together, so this paper fills a gap in the theory literature for these algorithms. Bounds are provided for both score-based methods and the more general predictor/rejector setting.

**Strengths:**

- Relatively well-presented and explained theoretical results
- Fills a gap in the existing theory literature for learning to defer, especially in the multi-expert setting.

**Weaknesses:**

- The empirical results are missing baselines / comparisons to other surrogate methods in the literature.

- I'm unable to follow the proofs of Theorems 5 and 7, which seem to be missing several steps (perhaps they're obvious?). In any case, the conclusions of these theorems seem to contradict the following simple 2D counterexample based on XOR.
Suppose $\mathcal{D}$ is uniform over $[-1,1] \times [-1,1]$. Set the label $Y=1$ when $X_1 \cdot X_2 \ge 0$, and set $Y=-1$ elsewhere (i.e., the label is positive in quadrants 1 and 3, and negative in quadrants 2 and 4). Suppose there is just one human expert, and the human is always correct when $X_1 < 0$ and always incorrect when $X_1 \ge 0$. This is clearly $(\mathcal{H},\mathcal{R})$-realizable for $\mathcal{H}$ and $\mathcal{R}$ both linear: $r^*$ chooses the human whenever $X_1 < 0$, and $h^*$ predicts $\mathrm{sign}(X_2)$, which is the correct classification whenever $X_1 \ge 0$.
However, the conclusion of the theorem seems false in this case. There is no halfspace that achieves a nontrivial accuracy on all of $\mathcal{D}$. However, we can still find an $h$ where $\varepsilon_{l_1}(h) - \varepsilon^*_{l_1}(\mathcal{H})$ is arbitrarily small. For a fixed $h$ (which will have a poor loss $l_1(h)$), we can still fit an optimal deferrer, i.e., we can still reduce $\varepsilon_{L_h}(r) - \varepsilon^*_{L_h}(\mathcal{R})$ towards 0. In this case, the RHS of the theorem statement can approach 0, but clearly, this combination has a poor loss $L_{def}$.
Essentially, the issue is that the right-hand-side of the theorem statement is not sensitive to whether two-stage optimization is suboptimal, but $(\mathcal{H}, \mathcal{R})$-realizability allows for the case where it is.

- Without the above, the results are still fairly interesting, since they provide some theory for two-stage methods, but the paper can't claim to resolve the open problem of Mozannar et al. 2023.

**Questions:**

- If I'm mistaken about the above (if so, I will increase my score), how do the results of Theorem 5 and 7 avoid examples where two-stage optimization is very suboptimal and joint optimization is required, like the one I presented above? Can the authors provide more verbose and rigorous proofs of Theorems 5 and 7?

- It might be useful to see if/how the proposed surrogate reduces to other ones in the literature in the single-expert case, depending on the choices of $l$.

**Limitations:**

I could not find a discussion of the limitations of the work.

---

> ### Author Rebuttal · Authors · 2023-08-10
>
> Thank you for your insightful comments. We have carefully addressed all the questions raised. Please find our responses below.
>
> **1. The empirical results are missing baselines / comparisons to other surrogate methods in the literature.**
>
> **Response:** To the best of our knowledge, our work is the pioneer in examining a two-stage scenario for learning to defer—a framework we argue is crucial in practice for many applications. Presently, we're unaware of any established baselines within this context. While the main focus of this work is a theoretical analysis, in the final version, we are happy to compare our novel surrogate losses with the existing single-stage surrogate losses in the literature, thereby enriching our experiments in line with the reviewer's suggestions.
>
> **2. Theorems 5 and 7.**
>
> **Response:** Thank you for your careful comments. Theorems 5 and 7 are analyzed under the assumption that the cost functions remain constant, thus $\alpha_j = 0$ in the cost function $c_j(x, y) = \alpha_j 1_{\mathsf h_j(x) \neq y} + \beta_j$. Regrettably, we omitted to indicate this assumption in the statements, and we apologize for the confusion this has caused. In contrast, our analysis for Theorem 1, Corollary 2, Theorem 6 is general and only requires the cost to be bounded. We will clarify these assumptions of our main results in the final version.
>
> Here is an alternative proof for the realizable consistency of our predictor-rejector two-stage surrogates with constant costs $c_j$, following the realizable consistency proof format in the recent work by Mozannar et al. [2023, Appendix D.4]. A similar proof applies to our score-based two-stage surrogates. Please let us know if these are sufficiently verbose as requested by the reviewer and if more clarification is needed.
>
> -------------------------------------------------------------------------
> Assume that $H$ and $R$ are closed under scaling. Let $\ell_1$ and $\ell_2$
> be the logistic loss.
> First, by definition, it is straightforward to see that for any $h,r,x,y$, $\mathsf L_{h}(r,x,y)$ upper bounds the deferral loss $\mathsf L_{\text{def}}(h,r,x,y)$. Consider a data distribution and costs under which there exists $h^*\in H$ and $r^*\in R$ such that $\mathcal{E} _ {\mathsf L _ {\text{def}}}(h^*,r^*)=0$.
>
> Let $\hat h$ be the minimizer of $\mathcal{E} _ {\ell_1}$ and $\hat r$ the minimizer of $\mathcal{E} _ {\mathsf L_{\hat h}}$
> Then, using the fact that $\mathsf L_{h}$ upper bounds the deferral loss $\mathsf L_{\text{def}}$, we have $\mathcal{E} _ {\mathsf L _ {\text{def}}}(\hat h, \hat r)\leq \mathcal{E} _ {\mathsf L_{\hat h}}(\hat r)$.
>
> Next we analyze two cases. If for a point $x$, deferral occurs, that is there exists $j^*\in [n_e]$, such that $\mathsf r^*(x)=j^*$, then we must have $c_{j^*} = 0$ for all $x$ since the data is realizable and $c_{j^*}$ is constant. Therefore, there exists an optimal $r^{\*\*}$ deferring all the points to the $j^*$th expert. Then, by the assumption that $R$ is closed under scaling and the Lebesgue dominated convergence theorem, for $\ell_2$ being the logistic loss, $\mathcal{E} _ {\mathsf L _ {\text{def}}}(\hat h, \hat r)\leq\mathcal{E} _ {\mathsf L _ {\hat h}}(\hat r) \leq \lim _ {\tau \to +\infty} \mathcal{E} _ {\mathsf L _ {\hat h}}(\tau r^{**})=0$, where we used the fact that in the limit of $\tau \to +\infty$ the logistic loss term $\ell_2(\overline r^{\*\*},x,j)$ corresponding to $j \neq j^*$ is zero.
>
> On the other hand, if no deferral occurs for any points, that is $\mathsf r^*(x)=0$ for any $x$, then we must have $1_{\mathsf h^*(x) \neq y}=0$ for all $(x,y)$ since the data is realizable. Using the fact that $H$ is closed under scaling and that the logistic loss is realizable $H$-consistent in the standard classification, we obtain $1_{\hat{\mathsf h}(x) \neq y}=0$ for all $(x,y)$. Then, by the assumption that $R$ is closed under scaling and the Lebesgue dominated convergence theorem, for $\ell_2$ being the logistic loss, $\mathcal{E} _ {\mathsf L _ {\text{def}}}(\hat h, \hat r)\leq\mathcal{E} _ {\mathsf L _ {\hat h}}(\hat r) \leq \lim_{\tau \to+\infty} \mathcal{E} _ {\mathsf L _ {\hat h}}(\tau r^{*})=0$, where we used the fact that in the limit of $\tau \to +\infty$ the logistic loss term $\ell_2(\overline r^{\*},x,j)$ corresponding to $j \neq 0$ is zero.
>
> Therefore, the optimal solution from minimizing predictor-rejector two-stage surrogates leads to a zero error solution of the deferral loss, which proves that the predictor-rejector two-stage surrogate loss is realizable consistent.
>
> -------------------------------------------------------------------------
>
> We concur, however, that for a more general cost function where $\alpha_j$ is not zero, our two-stage surrogate losses may lack realizable consistency, a point well illustrated by your example. Furthermore, we acknowledge that, in that respect, our claim about resolving the open problem from (Mozannar et al., 2023) is overstated and we will rectify that in the final version.
>
> Nonetheless, as the reviewer rightly notes, even if our realizable consistency results don't extend to the general cost function scenarios, we remain confident in the novelty and significance of our results related to two-stage learning to defer with multiple experts.
>
> **3. It might be useful to see if/how the proposed surrogate reduces ... choices of $\ell$.**
>
> **Response:** To the best of our knowledge, even in the context of the single-expert scenario, our family of two-stage surrogate losses are novel. However, we concur that showcasing the single-expert case can illuminate the uniqueness of our formulation. We will certainly incorporate a detailed discussion on this in the final version.
>
>
> **Limitations: I could not find a discussion of the limitations of the work.**
>
> **Response:** Thank you for pointing it out, we will make sure to add a separate discussion on the limitation in the final version.

---

> > ### Comment · Reviewer_Jgi5 · 2023-08-18
> > **update**
> >
> > Thanks to the authors for addressing my concerns. I’ve increased my score slightly but I think the paper remains very borderline because large changes to the framing and writing are required given that the proposed method doesn’t resolve the open problem of Mozannar et al., and the main contribution is the two-stage theory. For missing empirical comparisons, the most relevant one that comes to mind is Okati et al., which also trains the system in two phases.

---

> > > ### Author Response · Authors · 2023-08-19
> > > **Thank You**
> > >
> > > We appreciate the reviewer's response and are pleased that our rebuttal addressed their concerns, leading to an updated rating.
> > >
> > > Regarding writing, minor adjustments will clarify our work's connection to Mozannar et al.'s open problem. Our key contribution remains introducing principled surrogate losses for the deferral scenario, supported by $H$-consistency bounds, along with corresponding algorithms in score-based and predictor-rejector frameworks, and empirical results.
> > >
> > > We will reference (Okati et al., 2021) and emphasize differences in their learning scenario compared to ours, notably the need for conditional probabilities unavailable in our context, highlighted by Mozannar et al. (2023).

---

### Official Review · Reviewer_yH2K · 2023-07-04

**Soundness:** 4 excellent
**Presentation:** 3 good
**Contribution:** 3 good
**Rating:** 7
**Confidence:** 1

**Summary:**

This paper studies a two-stage scenario for learning to defer with multiple experts. For both the score-based setting and the predictor-rejector setting, the authors design a new family of surrogate loss function . The authors show that the surrogate losses are realizable H-consistent and  Bayes-consistency of our surrogate losses. The authors provide theoretical analysis and empirical results to justify the proposed framework.

**Strengths:**

1. The problem setup is interesting and realistic. The two stage deferring scenario is suitable when predictor is a pretrained LLM, and has promising real applications.

2. The surrogate loss function is well-motivated and new as far as I am concerned. The theoretical results are strong and comprehensive. I do not fully check the proof, but the results seem reasonable to me.

**Weaknesses:**

I am satisfied with the theoretical part. However, I would like to remark that I am not an expert in this field. I will be happy to follow up in the discussion period, if there is any additional questions raised up by other reviewers.

For the experiments, the accuracy on CIFAR10 and SVHN is significantly lower than the standard results of ResNet. Moreover, the authors only implement the proposed loss function. I would be better to compare with different baselines in previous works.

**Questions:**

See weakness part above.

**Limitations:**

I do not see the discussion of limitations, but I think this is not a major problem. I do not see potential negative societal impact as this paper mainly focuses on the theoretical side.

---

> ### Author Rebuttal · Authors · 2023-08-10
>
> Thank you for your appreciation of our work. We will take your suggestions into account when preparing the final version. Below please find responses to specific questions.
>
> **Weaknesses: I am satisfied with the theoretical part. However, I would like to remark that I am not an expert in this field. I will be happy to follow up in the discussion period, if there are any additional questions raised by other reviewers. For the experiments, the accuracy on CIFAR10 and SVHN is significantly lower than the standard results of ResNet. Moreover, the authors only implement the proposed loss function. It would be better to compare with different baselines in previous works.**
>
> **Response:** We did not aim to achieve high accuracy for each expert, which is not the focus of this paper. No data augmentation was used and the hyperparameters were not specifically tuned for each expert in our experiments. In the final version, we can report results for experts with higher accuracy.
>
> To the best of our knowledge, our work is the pioneer in examining a two-stage scenario for learning to defer—a framework we argue is crucial in practice for many applications. Presently, we're unaware of any established baselines within this context. While the main focus of this work is a theoretical analysis, in the final version, we are happy to compare our novel surrogate losses with the existing single-stage surrogate losses in the literature, thereby enriching our experiments in line with the reviewer's suggestions.
>
> **Limitations: I do not see the discussion of limitations, but I think this is not a major problem. I do not see potential negative societal impact as this paper mainly focuses on the theoretical side.**
>
> **Response:** Thank you for pointing it out, we will make sure to add a separate discussion on the limitation in the final version.

---

> > ### Comment · Reviewer_yH2K · 2023-08-17
> >
> > Thanks for the clarifications. I will keep my score.

---

### Official Review · Reviewer_Z2p9 · 2023-07-05

**Soundness:** 3 good
**Presentation:** 3 good
**Contribution:** 3 good
**Rating:** 7
**Confidence:** 2

**Summary:**

This paper provides bounds on the difference between the target loss and its surrogate loss, for two-stage learning to defer to multiple experts. The bounds guarantee that a small surrogate risk leads to a small target risk. The authors also show that in realizable cases, achieving the minimum surrogate risk results in the zero target risk, which affirmatively answers an open question. The paper studies two setups of two-stage learning to defer with multiple experts and provides similar theoretical results for both. Finally, the paper presents experiments using CIFAR-10 and SVHN, showing that having more experts improves performance.

**Strengths:**

- The paper studies the practically important task of learning to defer to experts.
- The results seem to be non-trivial and critical theoretical guarantees for the approach studied.
- The writing is overall well-written and easy to follow, except for some technical parts.
- The paper provides a good review of previous work and clearly explains the contributions.
- The experiment is interesting and shows the usefulness of the approach of learning to defer.

**Weaknesses:**

- Some technical parts were a little unclear. (See the questions below.)
- Section 3 and Section 4 are repetitive. I don't know how much the setups are different from each other.
- I think some math expressions could be simplified. (See the Questions section below.)
- The experiments are not extensive. I expected to see some experiments confirming the theoretical results, but the focus of the experiments presented in the paper seems to be the improvement by increasing the number of experts, which is quite independent of the theory part.
- The paper stresses the novelty of the proof technique and there are brief explanations, but it is difficult for me to understand them without reading the appendix. (Unfortunately, I did not have time to read the appendix.)

**Questions:**

- In line 21, the paper says, "However, this is not relevant in practice: the cost of retraining the predictor would be prohibitive for many applications." I think saying it is "not relevant" is a bit too strong if there are still some scenarios to which it is applicable.
- For the minimizability gap $\mathcal{R}^*_l - \mathbb{E}\_{x}[\inf\_{h\in \mathcal{H}} \mathbb{E}\_{y | x}[\texttt{L}(h, x, y)]]$, why do we want to consider the point-wise infimum for the second term? I think it is more natural to consider $\inf\_{h\in \mathcal{H}} \mathbb{E}\_{x, y}[\texttt{L}(h, x, y)]$.
- In lines 137-140, it is strange to "decompose" $\mathcal{H}$ as $\mathcal{H} = \mathcal{H}_p \times \mathcal{H}_d$ when $\mathcal{H}$ is a set of functions. ($\mathcal{H}_p \times \mathcal{H}_d$ is a set of function pairs.)
- I think it's nicer to express, e.g., Eq. (3) as $L_{h_p}(h_d, x, y) = \sum\_{j=0}^{n_e} c\_j(x, y) \ell_2(\bar{h}\_d, x, j)$, with $c_j$ extended by $c_0(x, y) = \mathbb 1\_{h\_p(x) \neq y}$. Equations are unnecessarily long and hard to read especially in Table 2.
- Is Definition 4 correct? (I think there is something wrong with the organization of the sentences.)
- What do the symbols in the "Base cost" column in Table 4 mean?
- What does Accuracy in Table 4 mean? They are not related to the deferral loss?
- In ll. 231-234, I could not understand this part: "In their proof, the authors choose $\bar h\_{\mu}$ based on individual scores $h(x, y)$, rather than the softmax. Consequently, if $h(x)$ or the label $y_{\text{max}}$ that achieves the maximum conditional probability corresponds to the label zero, $\mu$ cannot be optimized since the score $h(x, 0)$ is fixed at $\lambda$."

**Limitations:**

This paper focuses on bounds regarding population quantities. The results do not directly apply to empirical risk minimization although this extension does not seem very hard.

---

> ### Author Rebuttal · Authors · 2023-08-10
>
> Thank you for your appreciation of our work. We will take your suggestions into account when preparing the final version. Below please find responses to specific questions.
>
> **Weaknesses:**
>
> **1. Some technical parts were a little unclear. (See the questions below.)**
>
> **Response:** Thank you for highlighting those points. We value your feedback and will certainly consider your suggestions. Below, you'll find our detailed responses to your specific questions.
>
> **2. Section 3 and Section 4 are repetitive. I don't know how much the setups are different from each other.**
>
> **Response:** Section 3 delves into learning with deferral within a score-based framework, wherein the deferral is associated with extra scores. In contrast, Section 4 explores learning with deferral in the predictor-rejector setting, with the deferral corresponding to a separate function. These represent two separate learning frameworks that have been studied in the literature. Historically, for traditional single-stage scenarios, deriving consistent surrogate losses in the predictor-rejector setting has been challenging, leading many to opt for the score-based approach. We will further clarify this distinction and provide a detailed explanation in the final version.
>
> **3. I think some math expressions could be simplified. (See the Questions section below.)**
>
> **Response:** Thank you for highlighting those points. We will take into account your suggestions to refine and streamline the expressions. Below, you'll find our detailed responses to your specific questions.
>
> **4. The experiments are not extensive. I expected to see some experiments confirming the theoretical results, but the focus of the experiments presented in the paper seems to be the improvement by increasing the number of experts, which is quite independent of the theory part.**
>
> **Response:** Thank you for the suggestion. While the main focus of this work is a theoretical analysis of a two-stage scenario for learning to defer, we will seek to present a more comprehensive set of experiments in the final version. Specifically, we intend to incorporate comparisons with other single-stage surrogate losses in the literature, which has also been suggested by other reviewers.
>
> **5. The paper stresses the novelty of the proof technique and there are brief explanations, but it is difficult for me to understand them without reading the appendix. (Unfortunately, I did not have time to read the appendix.)**
>
> **Response:** Thank you for the suggestion. We will leverage the extra page allowed for the final version to include a more detailed discussion of the technical novelty in the main body.
>
> **Questions:**
>
> **1. In line 21 ...**
>
> **Response:** Thank you for the suggestion, we will refine it in the final version.
>
> **2. For the minimizability gap ...**
>
> **Response:** The first term $\mathcal{R}^*_{\mathsf L}(H)$ is the infimum of the expectation loss $\inf_{h \in H} \mathbb{E}_{x,y} [\mathsf L(h,x,y) ]$. Thus, the minimizability gap measures the difference of the best-in-class expected loss and that of the expected pointwise infimum loss. We will clarify it in the final version.
>
> **3. In lines 137-140 ...**
>
> **Response:** The decomposition here means that any function $h$ within $H$ is composed of the first $n$ scores $h_p$, used for prediction, alongside the final $n_{e}$ scores $h_d$, used for deferral. Thus, any $h\in H$ can be written as a pair $h=(h_p, h_d)$. We will further clarify that in the final version.
>
> **4. I think it's nicer to express ...**
>
> **Response:** Thank you for the feedback. We will enhance the presentation in line with your suggestions for the final version.
>
> **5. Is Definition 4 correct ...**
>
> **Response:** Sorry for the confusion. We will restructure the sentences in the final version. In short, Definition 4 says that if a surrogate loss is realizable $H$-consistent if, optimizing the surrogate loss results in the minimization of the deferral loss for any $H$-realizable distribution.
>
> **6. What do the symbols ...**
>
> **Response:** A check mark indicates the presence of a base cost in the cost function, whereas a cross mark signifies its absence. This will be made clearer in the final version.
>
> **7. What does Accuracy ...**
>
> **Response:** The accuracy refers to the overall accuracy of the learned pairs of predictor and deferral model. It is related to the deferral loss. Specifically, in the absence of the base cost, the accuracy aligns precisely with one minus the expected deferral loss.
>
> **8. In ll. 231-234 ...**
>
> **Response:** This mainly highlights that their method is tailored for hypothesis sets where each score can span across $\mathbb{R}$. This is not applicable in our context where the hypothesis set adheres to a predefined scoring function. In their proof, to set an upper bound on the estimation error of the zero-one loss using that of the surrogate loss, they select an auxiliary function $\overline{h}_{\mu}$ for any hypothesis $h$. This function is contingent on the distinct scores of $h$. Subsequently, the authors choose an optimal $\mu$ to set these bounds. Nevertheless, if any of $h$'s scores are fixed, an optimal $\mu$ doesn't exist, preventing the establishment of a meaningful bound. We will further clarify and detail these explanations in the final version.
>
> **Limitations: This paper focuses on bounds regarding population quantities. The results do not directly apply to empirical risk minimization although this extension does not seem very hard.**
>
> **Response:** Indeed, while our primary focus is on the infinite sample bound, our $H$-consistency bounds can be leveraged to derive finite sample learning bounds for a hypothesis set $H$. This can be achieved by upper bounding the estimation error of the surrogate loss using the standard Rademacher complexity bound associated with the hypothesis set and the loss function. We will add these to the final version.

---

> > ### Comment · Reviewer_Z2p9 · 2023-08-15
> > **Re: Rebuttal by Authors**
> >
> > I appreciate the authors' thorough response to my comments. I don't have further questions. I hope that the authors incorporate the suggested clarification improvements into the final version as promised.

---

### Official Review · Reviewer_fJtW · 2023-07-08

**Soundness:** 3 good
**Presentation:** 3 good
**Contribution:** 3 good
**Rating:** 7
**Confidence:** 4

**Summary:**

The paper proposes two stage learning algorithms for learning to defer framework. Learning to defer framework consists of two components, a rejector and a classifier. Previous literature has focussed on jointly training both the classifier and the rejector. This paper, instead, focuses on the two stage setup in designing learning to defer systems where a pre-trained classifier is provided and the rejector is to be trained in the second stage. This is a promising and more practical use case scenario of the deferral systems. To this end, the paper makes a series of contributions: 1. a simple and elegant formulation of the the said two stage setup 2. Proposing a family of surrogate loss functions for this setup as well as providing risk control guarantees based on $\mathcal{H}$-consistency bounds that builds upon the consistency bounds for the loss employed in both the stages. 3. The paper then prove a more general result on the $\mathcal{H}$ consistency bounds for general surrogate losses for classification problem which is augmented with some additional instance dependent hypothesis. 4. The paper uses this result to establish $\mathcal{H}$ consistency as well as the Bayes consistency of the proposed two stage surrogates. Additionally, the paper also establishes the realizable $\mathcal{H}$ consistency of the two stage surrogates which is a more stronger property. The focus of the paper is general where multiple experts are possible.

**Strengths:**

1. The paper investigates a question of practical relevance to the machine learning community. Deferral systems have appealing properties as they foster trust and transparency, as well as monitoring behaviour by introducing experts into the system. However, it was not possible based on the prior work how to adapt any pre-trained system to experts. This paper fills this gap by devising theoretically sound algorithms.

2. While the paper builds heavily on the previous works on rejection learning, the proposed algorithms are sufficiently novel. The paper also fills a gap in the literature for deriving $\bar{\mathcal{H}}$-consistency bounds for a general rejection learning problem. As a plus point, the proposed surrogates are simple and intuitive which makes it easier for adoption in the community.

3. The paper proposes first of its kind $\mathcal{H}$-realizable consistent algorithms for learning to defer with multiple experts. I believe this is a significant contribution as generally building deferral systems considering multiple experts from scratch would be prohibitively expensive.

4. Although there are some minor typos in the text, the paper is clearly written and is super easy to read.



**Weaknesses:**

1. While theoretical results is the main focus of this paper, I believe experimental section could have been more insightful. Currently, the experimental section says that the overall system performance gets better on inclusion of more experts. However, this is intuitive and already known. Could authors add one more column in Table 4 to mention the performance of the originally trained classifier? I am curious to learn how the second stage adoption affects the originally trained classifier. While one can lower the estimation error of the first classifier by $\epsilon_{1}$, it is not clear (to me) where such bound still holds True after the adoption in the second stage. I agree that this is certainly not the focus of the paper, I believe it is important to know if this second-stage training degrades the already trained classifier. I assume not, right?

2. This might be a misunderstanding on my side, but I believe the proposed formulation can only work when $\alpha_{j}, \beta_{j} > 0,  \forall j$ (line 104). However, there are no guidelines on how to define these two quantities, or what is their nature.

3. One of the most promising appeal of the two stage training is preventing costly design from scratch. However, it is not clear whether it also has some disadvantages or not. Consider a single stage training approach where one wants to bound the estimation error by $\epsilon$? Would it be cheaper / expensive to achieve the same bound using the two-stage approach? Could some quantitative results be given to this end to guide the practitioners?

**Questions:**

1. $\mathcal{H}$ realizable consistent surrogates have not been proposed in the literature for single stage learning to defer with multiple experts. Is it possible to utilise the results in this paper for that? For example, maybe using iterative training procedure.

**Limitations:**

No glaring limitations.

---

> ### Author Rebuttal · Authors · 2023-08-10
>
> Thank you for your encouraging review. We will take your suggestions into account when preparing the final version. Below please find responses to specific questions.
>
> **Weaknesses:**
>
> **1. While theoretical results is the main focus of this paper, I believe experimental section could have been more insightful. Currently, the experimental section says that the overall system performance gets better on inclusion of more experts. However, this is intuitive and already known. Could authors add one more column in Table 4 to mention the performance of the originally trained classifier? I am curious to learn how the second stage adoption affects the originally trained classifier. While one can lower the estimation error of the first classifier by $\epsilon_1$, it is not clear (to me) where such bound still holds True after the adoption in the second stage. I agree that this is certainly not the focus of the paper, I believe it is important to know if this second-stage training degrades the already trained classifier. I assume not, right?**
>
> **Response:** The accuracy of the originally trained classifier on SVHN is 91.12% and that on CIFAR-10 is 70.56%. We will add them to Table 4 in the final version.
>
> That’s right. The second-stage training does not degrade the performance of the already trained classifier.
>
> **2. This might be a misunderstanding on my side, but I believe the proposed formulation can only work when $\alpha_j, \beta_j>0, \forall j$ (line 104). However, there are no guidelines on how to define these two quantities, or what is their nature.**
>
> **Response:** Our analysis for Theorem 1, Corollary 2, Theorem 6 is general and only requires the cost to be bounded. This formulation is one natural choice to define the cost. Typically, the hyperparameter $\alpha_j$ has two potential values: zero or one. When $\alpha_j$ is set to one, the first term of the formulation pertains to the inaccuracy of expert expert $h_j$. Conversely, with $\alpha_j$ set to zero, the first term vanishes, focusing solely on the inference cost. Theorems 5 and 7 are analyzed under this assumption. The $\beta_j$ in the second term, for example, can correspond to the inference cost incurred by expert $h_j$. We will further clarify and elaborate on these in the final version.
>
> **3. One of the most promising appeal of the two stage training is preventing costly design from scratch. However, it is not clear whether it also has some disadvantages or not. Consider a single stage training approach where one wants to bound the estimation error by $\epsilon$? Would it be cheaper / expensive to achieve the same bound using the two-stage approach? Could some quantitative results be given to this end to guide the practitioners?**
>
> **Response:** Great question! Intuitively, the drawback of a two-stage approach lies in the potential accumulation of errors across both the first and second stages. In contrast, the single-stage approach sidesteps this problem, allowing for a more streamlined optimization. However, in some cases, a direct single-stage loss might be harder to optimize computationally. A finer theoretical comparison between single-stage and two-stage approaches would certainly be interesting and we intend to further investigate that.
>
> **Questions:**
>
> **1. $\mathcal{H}$-realizable consistent surrogates have not been proposed in the literature for single stage learning to defer with multiple experts. Is it possible to utilise the results in this paper for that? For example, maybe using iterative training procedure.**
>
> **Response:** Great question! This is in fact a research direction we've recently embarked upon exploring.

---

> > ### Comment · Reviewer_fJtW · 2023-08-15
> > **Post rebuttal comment**
> >
> > Thanks to authors for the response. It answers my questions. I'd suggest authors to elaborate more on Q2 in the main text, along with the points raised by reviewer Jgi5. I do not have anymore questions, and I'm happy to increase my score as I believe the paper should appear at NeurIPS.

---

### Decision · Program_Chairs · 2023-09-21

**Decision:**

Accept (poster)

**Comment:**

This paper analyzes the realizability of two-stage learning-to-defer algorithms, providing new insights to previous results that suggested the superiority of one-stage / joint algorithms.  The reviewers are unanimously in favor of acceptance.  The most critical stance was presented by Reviewer Jgi5, who questioned the realizability of the consistent surrogates in cases and claimed the paper overstates its resolution of the the open problem from (Mozannar et al., 2023).  While this critique has merits, I find the authors successfully rebutted the former theoretical point.  For the latter point, the authors should be sure to temper their claim about resolving the open problem(s) from (Mozannar et al., 2023) in the camera-ready version, as they promised to do in their reply to Reviewer Jgi5.